# INSTANTPORTRAIT: ONE-STEP PORTRAIT EDITING VIA DIFFUSION MULTI-OBJECTIVE DISTILLATION

**Zhixin Lai**[1*]     **Keqiang Sun**[2]     **FuYun Wang**[2]     **Dhritiman Sagar**[1]     **Erli Ding**[1*]

[1] Snap Inc     [2] The Chinese University of Hong Kong

{laizhixin16,erliding}@gmail.com, {kqsun,fywang}@link.cuhk.edu.hk, {dhritiman.sagar}@snapchat.com

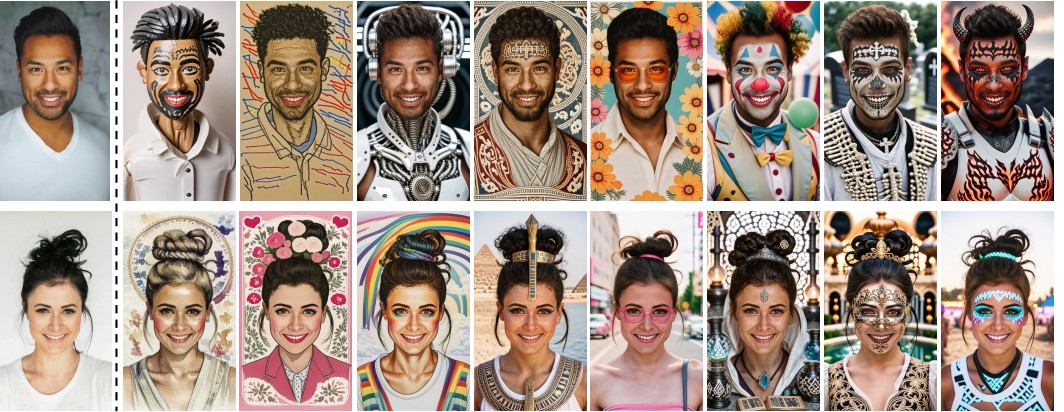

Input     Output images conditioned by the input images and editing instructions

Figure 1: The output of Instant-Portrait model (IPNet). IPNet is a portrait image editing model trained through Diffusion Multi-Objective Distillation. IPNet excels in identity preservation and portrait editing, while also achieves one-step model inference. (Prompts are in Appendix I)

## ABSTRACT

Real-time instruction-based portrait image editing is crucial in various applications, including filters, augmented reality, and video communications, *etc.* However, real-time portrait editing presents three significant challenges: identity preservation, fidelity to editing instructions, and fast model inference. Given that these aspects often present a trade-off, concurrently addressing them poses an even greater challenge. While diffusion-based image editing methods have shown promising capabilities in personalized image editing in recent years, they lack a dedicated focus on portrait editing and thus suffer from the aforementioned problems as well. To address the gap, this paper introduces an Instant-Portrait Network (IPNet), the first one-step diffusion-based model for portrait editing. We train the network in two stages. We first employ an annealing identity loss to train an Identity Enhancement Network (IDE-Net), to ensure robust identity preservation. We then train the IPNet using a novel diffusion Multi-Objective Distillation approach that integrates adversarial loss, identity distillation loss, and a novel Facial-Style Enhancing loss. The Diffusion Multi-Objective Distillation approach efficiently reduces inference steps, ensures identity consistency, and enhances the precision of instruction-based editing. Extensive comparison with prior models demonstrates IPNet as a superior model in terms of identity preservation, text fidelity, and inference speed.

---

[*]Equal contribution. Work done while at Snap Inc.

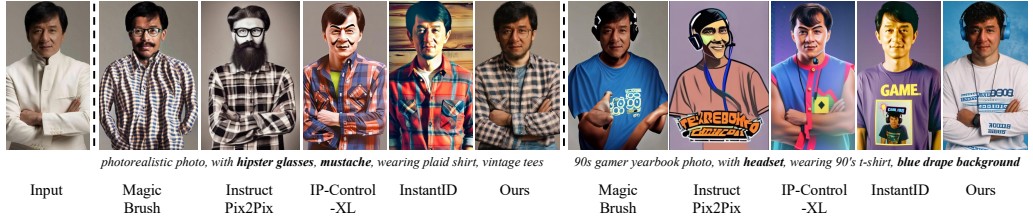

*photorealistic photo, with **hipster glasses**, **mustache**, wearing plaid shirt, vintage tees*     *90s gamer yearbook photo, with **headset**, wearing 90's t-shirt, **blue drape background***

| Input | Magic Brush | Instruct Pix2Pix | IP-Control -XL | InstantID | Ours | Magic Brush | Instruct Pix2Pix | IP-Control -XL | InstantID | Ours |

Figure 2: Bad cases of SOTA methods. Image editing models like MagicBrush Zhang et al. (2024) and InstructPix2Pix Brooks et al. (2023) frequently cause significant identity distortion and artifacts in complex edits. In identity preservation modes like IP-Adapter Ye et al. (2023) with ControlNet Zhang et al. (2023) SD-XL (IP-Control-XL) and InstantID Wang et al. (2024a), identity is better maintained but style effects and text fidelity are bad. Simultaneously excelling in both identity preservation and text fidelity remains a challenge.

# 1 INTRODUCTION

Instruction-based portrait image editing is defined as maintaining the face identity while allowing modifications to all other aspects such as background, clothing, and facial features. Instruction-based portrait image editing is critical for applications such as digital lens filters, augmented reality, and video communications.

For evaluation of the portrait image editing models, people mainly care about three aspects: the identity consistency Tewari et al. (2020); Sun et al. (2023), the editing fidelity Gu et al. (2019); Xia et al. (2021), and the speed Portenier et al. (2018); Kim et al. (2021); Bai et al. (2024). These aspects are typically a trade-off Fitzsimmons et al. (2018); Alaluf et al. (2022); Li et al. (2023a), which poses a significant challenge to improve all of these aspects simultaneously.

Recent image editing models such as InstructPix2Pix Brooks et al. (2023) and MagicBrush Zhang et al. (2024) focus on text fidelity, enabling detailed edits based on user instructions but often struggle with maintaining consistent identity, resulting in significant discrepancies and distortions, particularly in complex portrait edits (see Figure 2). A primary reason for these issues is the lack of datasets that are both robust in identity preservation and rich in complex portrait edits, along with insufficient supervision of identity during training. On the other hand, personalized image synthesis models like IP-Adapter Ye et al. (2023) and InstantID Wang et al. (2024a) enhance identity preservation compared to image editing models but at the expense of style, particularly in facial effects, as shown in Figure 2. This trade-off results from using identity embeddings as an input condition, which limits the ability to edit facial style effects. Additionally, using identity embeddings as the sole input condition without identity-specific supervision has been proved to be an inadequate method for preserving identity, with noticeable discrepancies compared to the original identity as illustrated in Figure 2. Furthermore, the models require 30 - 50 sampling steps in inference.

To address these challenges, we first construct a dataset specifically designed for instruction-based portrait image editing. Using the dataset, we train our Identity Enhancement Network (IDE-Net) and the Instant-Portrait Network (IPNet) through a two-stage approach, optimizing for identity preservation, precise image editing, and inference speed:

(1) In the first stage, we train IDE-Net with a novel Annealing Identity Loss, which uses the input face embedding as supervision and employs an annealing strategy to dynamically balance the identity loss and stable diffusion loss. This approach enables our IDE-Net to generate high-quality images that maintain precise identity, even better than the training dataset.

(2) In the second stage, we distill IDE-Net into IPNet with our novel Diffusion Multi-Objective Distillation technique. During the distillation, we incorporate an Adversarial Loss to enhance the image quality, integrating Identity Distillation Loss to preserve the identity, and also utilizing a Face-Style Enhancing Triplet Loss to further enhance the facial style. IPNet

achieves faster inference, higher image quality, and better facial style, than the teacher model IDE-Net.

In our evaluation, IPNet is compared with top state-of-the-art models in portrait image editing across metrics including identity preservation, text fidelity, and image quality, as well as model size, and inference steps. IPNet outperforms these models by a substantial margin, demonstrating the effectiveness of our Diffusion Multi-Objective Distillation approach.

The core contributions are three folded:

(1) To our knowledge, we are the first to achieve one-step instruction-based portrait image editing, with high-precision identity preservation, precise instruction-based image editing, and rapid model inference concurrently.

(2) We introduce the *Constant Identity Loss* to significantly improve identity preservation. Building on this, we propose the *Annealing Identity Loss*, an enhancement of the *Constant Identity Loss* that dynamically adjusts to better balance identity preservation and text alignment during IDENet training.

(3) We introduce the *Diffusion Multi-Objective Distillation* process to distill IDENet into IPNet with *Adversarial Loss*, *Identity Distillation Loss*, and *Face-Style Enhancing Triplet Loss*, achieving one-step instruction-based portrait image editing and balancing multiple concerned objectives.

## 2 Preliminary

Stable Diffusion Kingma & Welling (2013) utilizes a pre-trained variational autoencoder with an encoder $\mathcal{E}$ and a decoder $\mathcal{D}$. For an image $x$, the encoder produces an initial latent $z = \mathcal{E}(x)$, and during the diffusion process, noise is incrementally added, creating a noisy latent $z_t$ at each timestep $t \in T$. The transformation for each step is mathematically described as follows:

$$z_t = \sqrt{\bar{\alpha}_t} z_0 + \sqrt{1 - \bar{\alpha}_t}\epsilon, \quad \epsilon \sim \mathcal{N}(0, I) \tag{1}$$

where $\bar{\alpha}_t$ is a fixed noise scaling factor, and $\epsilon$ is noise sampled from a standard normal distribution.

Building on the diffusion model, InstructPix2Pix Brooks et al. (2023) adapts a controlled diffusion framework for image editing. It employs a U-Net $\epsilon_\theta$ to predict the noise in the latent $z_t$, guided by image conditions $c_I$ and text instructions $c_T$. InstructPix2Pix modifies the U-Net's first convolutional layer to include additional channels that concatenate $c_I$ with $z_t$. The diffusion model loss $(L)_{\mathrm{dm}}$ of InstructPix2Pix is as follows:

$$\mathcal{L}_{\mathrm{dm}} = \mathbb{E}_{\mathcal{E}(x), \mathcal{E}(c_I), c_T, \epsilon, t} \left[ \| \epsilon - \epsilon_\theta(z_t, t, \mathcal{E}(c_I), c_T)) \|_2^2 \right] \tag{2}$$

For the instruction-driven portrait image editing, InstructPix2Pix requires precise identity alignment between input and target images for model training, otherwise it would result in identity distortion, as shown in Figure 2.

## 3 Method

Our objective is to swiftly edit portrait images that preserve identity and comply with specific instructions. We first construct a tailored dataset for the instruction-based portrait editing, described in Appendix B, and then develop a two-stage training approach for our diffusion models, Identity Enhancement Network (IDE-Net) and Instant-Portrait Network (IPNet). Initially, as outlined in Figure 3, the IDE-Net is trained using a novel Annealing Identity Loss $\mathcal{L}_{\mathrm{aid}}$ (Section 3.1.1) to ensure robust identity preservation, achieving performance beyond the dataset's baseline. Subsequently, as outlined in Figure 4, we initialize the IPNet by cloning IDE-Net and augmenting it with our Diffusion Multi-Objective Distillation technique. This technique integrates Adversarial Loss $\mathcal{L}_{\mathrm{adv}}$ (Section 3.2.1) for enhancing image quality,

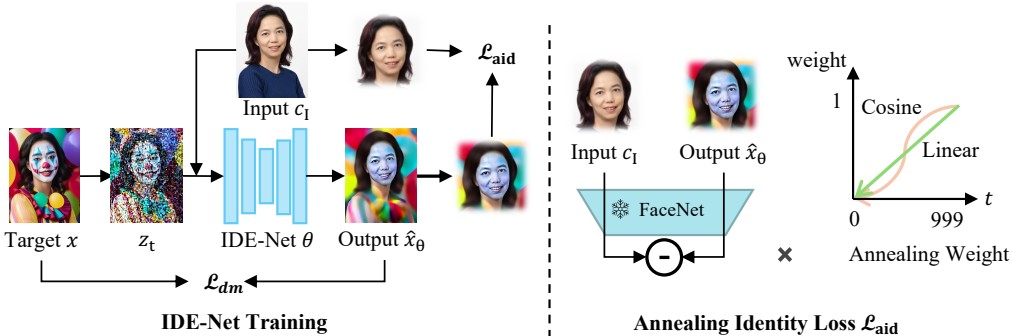

Figure 3: We train our Identity Enhancement Network (IDE-Net) with a novel Annealing Identity Loss to excel in identity preservation.

Identity Distillation Loss $\mathcal{L}_{\text{distill}}$ (Section 3.2.2) for maintaining identity, and Face-Style Enhancing Triplet Loss $\mathcal{L}_{\text{triplet}}$ (Section 3.2.3) for facial style improvement. As a result, IPNet achieves faster inference, superior image quality, and enhanced facial style than IDE-Net.

## 3.1 IDE-Net Training

For image editing conditioned by instruction text, InstructPix2Pix relies on precise identity alignment between input and target images in its training data, as discussed in Section 2. To address this constraint, we propose IDE-Net, built on InstructPix2Pix, incorporating a new Annealing Identity Loss $\mathcal{L}_{\text{aid}}$ that preserves identity more effectively, even with suboptimal datasets, as shown in Figure 3.

### 3.1.1 Annealing Identity Loss

To ensure identity preservation, we utilize an Constant Identity Loss $\mathcal{L}_{\text{cid}}$. This involves cropping facial regions from both the input image $c_I$ and the output image $\hat{x}_\theta$, which are then converted to grayscale $F_g$ to emphasize structural features over color and texture. Face embeddings are extracted using FaceNet $F_{\text{crop}}$ Schroff et al. (2015), and the $\mathcal{L}_{\text{cid}}$ is computed based on the $L_2$ distance between these embeddings, as detailed in Equation (3).

$$\mathcal{L}_{\text{cid}} = \mathbb{E}_{c_I, \hat{x}_\theta} \left[ \| F_{\text{crop}}(F_g(c_I)) - F_{\text{crop}}(F_g(\hat{x}_\theta)) \|_2^2 \right] \tag{3}$$

Prior works, such as Ju et al. (2023); Li et al. (2024), mention the importance of preserving face identity and pose structure in early denoising stages and enhancing style later. Therefore, to balance diffusion loss $L_{dm}$ and identity loss $L_{\text{id}}$, we introduce the Annealing Identity Loss $\mathcal{L}_{\text{aid}}$. This strategy involves gradually reducing the weight of the identity loss across the denoising steps, enabling a smooth transition from identity preservation to text alignment.

$$\mathcal{L}_{\text{aid}} = W_a(t, T_{max}) * \mathcal{L}_{\text{cid}} \tag{4}$$

where $T_{max}$ represents the maximum timestep in $T$ and usually sets 999 for stable diffusion models. $t$ denotes the timesteps, decreasing from $T_{max}$ to 0. The function $W_a(t, T_{max})$, an annealing algorithm, decreases as $t$ reduces, leading to a progressive reduction in the Annealing Identity Loss. Among various options, we opt for linear decay $\frac{t}{T_{max}}$, with comparative analysis in Appendix E.

The total loss for IDE-Net is defined in Equation (5). Initially, $L_{\text{IDE-Net}}$ effectively preserves the identity of the input image using $\mathcal{L}_{\text{aid}}$. As the denoising process progresses, $L_{\text{IDE-Net}}$ gradually shifts focus towards emphasizing style using $L_{\text{dm}}$, ensuring uniform enhancements throughout the denoising steps.

$$\mathcal{L}_{\text{IDE-Net}} = \mathcal{L}_{\text{dm}} + \lambda_{\text{aid}} * \mathcal{L}_{\text{aid}} \tag{5}$$

where $\lambda_{\text{aid}}$ is the balancing weight for the Annealing Identity Loss.

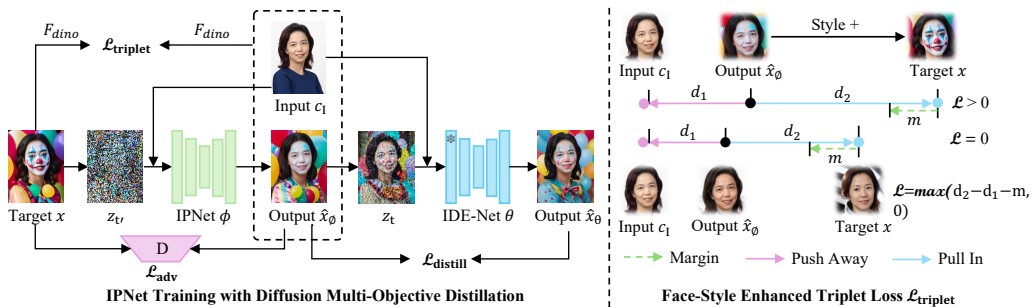

Figure 4: Our Instant-Portrait Network is distilled by IDE-Net with a novel Diffusion Multi-Objective Distillation technique. During distillation, we freeze the IDE-Net and incorporate an Adversarial Loss $\mathcal{L}_{\text{adv}}$ to improve image quality, an Identity Distillation Loss $\mathcal{L}_{\text{distill}}$ to maintain identity, and a Face-Style Enhancing Triplet Loss $\mathcal{L}_{\text{triplet}}$ to further refine facial style.

Training with $\mathcal{L}_{\text{aid}}$ results in a consistency of identity, but the style described by the text prompt is limited in the outputs of IDE-Net, which is then tackled by the introduced IPNet with Diffusion Multi-Objective Distillation.

## 3.2 IPNet Training with Diffusion Multi-Objective Distillation

As depicted in Figure 4, IPNet $\epsilon_\phi$, cloning from IDE-Net $\epsilon_\theta$, is trained with the Diffusion Multi-Objective Distillation method, inspired by the Score Distillation Sampling (SDS) Poole et al. (2022). The distillation process integrates an Adversarial Loss $\mathcal{L}_{\text{adv}}$ for enhancing image quality (Section 3.2.1), an Identity Distillation Loss $\mathcal{L}_{\text{distill}}$ to improve identity preservation (Section 3.2.2), and a Face-Style Enhancing Triplet Loss $\mathcal{L}_{\text{triplet}}$ to boost facial style editing capabilities (Section 3.2.3).

### 3.2.1 Adversarial Loss

The Adversarial Loss $\mathcal{L}_{\text{adv}}$ accelerates generation by enabling the model to learn complex transformations between marginal distributions Xiao et al. (2021), which can effectively reduce the number of denoising steps and can even achieve one-shot generation. Therefore, we incorporate a discriminator $D$, as suggested by Sauer et al. (2023a), which uses $\mathcal{L}_{\text{adv}}$ to distinguish between the output image of IDE-Net $\hat{x}_\phi$ (labeled as fake) and the target image $x$ (labeled as real).

In contrast to the text conditional discriminators referenced in Sauer et al. (2023a); Kang et al. (2023), we utilize an unconditional discriminator. This choice is supported by our findings that random cropping significantly enhances generative performance. However, such cropping can strip away crucial image semantics and adversely affect text conditions. To resolve this issue, we optimize the text alignment through knowledge distillation from the teacher model described in Section 3.2.2 rather than in the $\mathcal{L}_{\text{adv}}$. Therefore, the $\mathcal{L}_{\text{adv}}$ is formulated as:

$$\mathcal{L}_{\text{adv}} = \mathbb{E}_x\Big[\max(0, 1 - D(x))\Big] + \mathbb{E}_{\hat{x}_\phi}\Big[\max(0, 1 + D(\hat{x}_\phi))\Big] \tag{6}$$

### 3.2.2 Identity Distillation Loss

Score Distillation Sampling (SDS) Poole et al. (2022) is critical for Stable Diffusion model distillation Sauer et al. (2023b), as defined in Equation (10) and Equation (8):

$$\mathcal{L}_{\text{sds}} = \mathbb{E}_{z_t, \hat{x}_\phi}\Big[\|\hat{x}_\theta(stop\_grad(z_t)) - \hat{x}_\phi\|_2^2\Big] \tag{7}$$

$$z_t = \sqrt{\alpha_t}\hat{z}_\phi + \sqrt{1 - \alpha_t}\epsilon_t \tag{8}$$

where $\hat{x}_\phi$ is the output of IPNet with $\hat{z}_\phi$ being its corresponding predicted latent code. $z_t$, the input of IDE-Net, is calculated by applying the stochastic noise $\sqrt{1 - \alpha_t}\epsilon_t$ to $\sqrt{\alpha_t}\hat{z}_\phi$. $\hat{x}_\theta(stop\_grad(z_t))$ is the output of IDE-Net with a stop-gradient operation. The loss is defined as the $L2$ distance between the image output of IPNet and IDE-Net.

However, SDS often produces over-smoothed and low-detail images due to its reliance on stochastic noise sampling Wang et al. (2024b). To address this limitation, we integrate DDIM inversion Song et al. (2020) to the time steps lower than $\tau$ for fine-grained distillation. And for the time steps higher than $\tau$, which captures structure information, like pose and identity, we maintain the coarse-grained distillation with stochastic sampling, formulated as:

$$z_{t'} = \begin{cases} \sqrt{\alpha_t}\hat{z}_\phi + \sqrt{1 - \alpha_t}\hat{\epsilon}_\theta(\hat{z}_\phi, t) & t \le \tau \\ \sqrt{\alpha_t}\hat{z}_\phi + \sqrt{1 - \alpha_t}\epsilon_t & t > \tau \end{cases} \tag{9}$$

Here, the time-step threshold $\tau$ is a hyper-parameter, manually determined to 200 by checking the spatial alignment accuracy and image quality. When the time step $t > \tau$, we apply stochastic noise $\sqrt{1 - \alpha_t}\epsilon_t$ to $\sqrt{\alpha_t}\hat{z}_\phi$. When $t \le \tau$, DDIM inversion directly predicts $z_t$ by IDE-Net $\epsilon_\theta$, omitting stochastic noise. We call the new loss with the refined $z_{t'}$ as $\mathcal{L}_{\text{distill}}$.

$$\mathcal{L}_{\text{distill}} = \mathbb{E}_{z_t, \hat{x}_\phi}\left[\|\hat{x}_\theta(stop\_grad(z_{t'})) - \hat{x}_\phi\|_2^2\right] \tag{10}$$

### 3.2.3 FACE-STYLE ENHANCING TRIPLET LOSS

Preserving identity while updating facial style poses a significant challenge due to their interdependence. To address this, we introduce the Face-Style Enhancing Triplet Loss $\mathcal{L}_{\text{triplet}}$, which balances identity preservation and facial style variation by comparing the relations of input, output, and target images, as shown in Figure 4. The loss is outlined below.

$$\mathcal{L}_{\text{triplet}} = Max(d_2 - d_1 - m, 0) \tag{11}$$

where

$$d_1 = ||F_{\text{dino}}(F_{\text{crop}}(c_I)) - F_{\text{dino}}(F_{\text{crop}}(\hat{x}_\phi))||, \quad d_2 = ||F_{\text{dino}}(F_{\text{crop}}(x)) - F_{\text{dino}}(F_{\text{crop}}(\hat{x}_\phi))|| \tag{12}$$

Here, $c_I$ is the input, $\hat{x}_\phi$ is the output from IPNet, and $x$ is the target image. $F_{\text{crop}}$ denotes the facial crop, and $F_{\text{dino}}$ refers to the mapping of image to a $DINOv2$ embedding Oquab et al. (2023). $d_1$ represents the distance between the output and input embedding, and $d_2$ represents the $L_2$ distance between the output and target embedding. The margin $m$, tailored to datasets, serves as a dynamic mechanism to balance identity preservation with facial style. Typically, we set $m$ based on the observed distances in our validation set to ensure appropriate thresholds.

Overall, the total loss for IPNet is defined as:

$$\mathcal{L}_{\text{IPNet}} = \mathcal{L}_{\text{adv}} + \lambda_{\text{distill}} * \mathcal{L}_{\text{distill}} + \lambda_{\text{triplet}} * \mathcal{L}_{\text{triplet}} \tag{13}$$

where $\lambda_{\text{distill}}$ and $\lambda_{\text{triplet}}$ balances the loss functions, varying across different time-step sampling stages during distillation, as detailed in Section 4.1.

### 3.2.4 STYLE BOOST VIA ITERATIVE INFERENCE

To further improve the facial style, we apply the iterative inference by using the output of the first step in IPNet inference instead of the original image as the input for the second step. By feeding the output back into the model, subsequent steps further concentrate on stylistic details, amplifying the intended visual characteristics while preserving the overall structure.

Table 1: Quantitative comparison against state-of-the-art models. The best results are highlighted in **bold**, and the second-best results are underlined.

| Method | Image Resolution | Model Size↓ | Inference Step↓ | Face-Similarity↑ | | Text-Fidelity↑ | | Image-Quality↑ | | |
|---|---|---|---|---|---|---|---|---|---|---|
| | | | | FaceNet | InsightFace | CLIP-Vit-g | CLIP-Vit-H | HPS | Q-Align-Q | Q-Align-A |
| MagicBrush | 768x512 | 859M | 50 | 0.461 | 0.401 | 0.219 | 0.262 | 0.209 | 4.192 | 3.148 |
| InstructPix2Pix | 1024x576 | 859M | 30 | 0.263 | 0.196 | 0.247 | 0.287 | 0.216 | 3.653 | 2.755 |
| IP-Control-1.5 | 768x512 | 1852M | 30 | 0.794 | 0.670 | 0.235 | 0.284 | 0.247 | 4.546 | 3.160 |
| IP-Control-XL | 1024x576 | 3759M | 30 | 0.662 | 0.548 | 0.228 | 0.273 | 0.228 | 4.061 | 2.890 |
| InstantID | 1024x576 | 4150M | 50 | 0.751 | 0.689 | 0.257 | 0.296 | 0.263 | 4.022 | 3.087 |
| **IPNet (Ours)** | 1024x576 | 859M | 1 | **0.867** | **0.782** | **0.271** | **0.309** | **0.275** | **4.960** | **3.629** |

## 4 EXPERIMENTS

We outline the experiment setup in Section 4.1, compare IPNet with state-of-the-art models (SOTA) in Section 4.2, and conduct ablation studies in Section 4.3 to evaluate components' effectiveness.

### 4.1 EXPERIMENT SETUP

**Dataset** For training, we utilize our dataset outlined in Appendix B. For evaluation, we select 100 images from FFHQ Karras et al. (2019) and 40 prompts from our dataset, generating 4000 validation data pairs.

**Baseline** Our baseline model, InstructPix2Pix, utilizes Stable Diffusion V1.5 and is fine-tuned on our dataset.

**Evaluation metrics** We evaluate models along four dimensions: Speed, Face-Similarity, Text-Fidelity, and Image-Quality. *Speed* metrics include model size, and inference steps. *Face-Similarity* measures similarity between the output and the input image using FaceNet Schroff et al. (2015) and InsightFace Guo & Deng (2019). *Text-Fidelity* involves CLIP-T scores (CLIP-Vit-g and CLIP-Vit-H) Hessel et al. (2021), which evaluate semantic similarity between the output image and the condition text. Lastly, *Image-Quality* is assessed through Human Preference Score (HPS) Wu et al. (2023b), Q-Align-Quality (Q-Align-Q), and Q-Align-Aesthetic (Q-Align-A) Wu et al. (2023a), focusing on image quality and aesthetics.

**Implementation** IDE-Net is trained with time steps sampled in the range [0, 999], using an Annealing Identity Loss $\mathcal{L}_{\text{aid}}$ weight of 0.7 to balance identity preservation and image quality. Building on IDE-Net, IPNet distillation is divided into three stages: High Time Step ([400, 800]) focuses on the structure and pose alignment with Identity Distillation Loss $\mathcal{L}_{\text{distill}}$ set to 1 to stabilize early training; Middle Time Step ([200, 400]) reduces Identity Distillation Loss $\mathcal{L}_{\text{distill}}$ weight to 0.3, emphasizing Adversarial Loss $\mathcal{L}_{\text{adv}}$ to improve image quality and mitigate artifacts and further keeping structure and pose alignment; and Low Time Step ([150, 200]) refines details and style with DDIM inversion-based Identity Distillation Loss $\mathcal{L}_{\text{distill}}$ and Face-Style Enhancing Triplet Loss $\mathcal{L}_{\text{triplet}}$, utilizing a large batch size of 2048 to enhance style consistency and stabilize training. More training details are summarized in Appendix G.

### 4.2 BASELINE COMPARISON

To demonstrate the effectiveness of our method, we compare our IPNet with state-of-the-art (SOTA) models on portrait image editing, including MagicBrush Zhang et al. (2024), InstantID Wang et al. (2024a), IP-Adapter Ye et al. (2023) with ControlNet Zhang et al. (2023) SD1.5 (IP-Control-1.5), IP-Adapter with ControlNet SDXL (IP-Control-XL), and InstructPix2Pix SDXL Brooks et al. (2023). The integration of IP-Adapter with ControlNet aims to preserve the pose of the input image.

**Qualitative comparison** As shown in Figure 5, we assessed our IPNet models using diverse images and instruction prompts. The results confirm that IPNet surpasses SOTA models in Face-Similarity, Text-Fidelity, and Image-Quality.

**Quantitative comparison** As reported in Table 1, IPNet excels across Face-Similarity, Text-Fidelity, and Image-Quality metrics with fewer inference steps compared to other

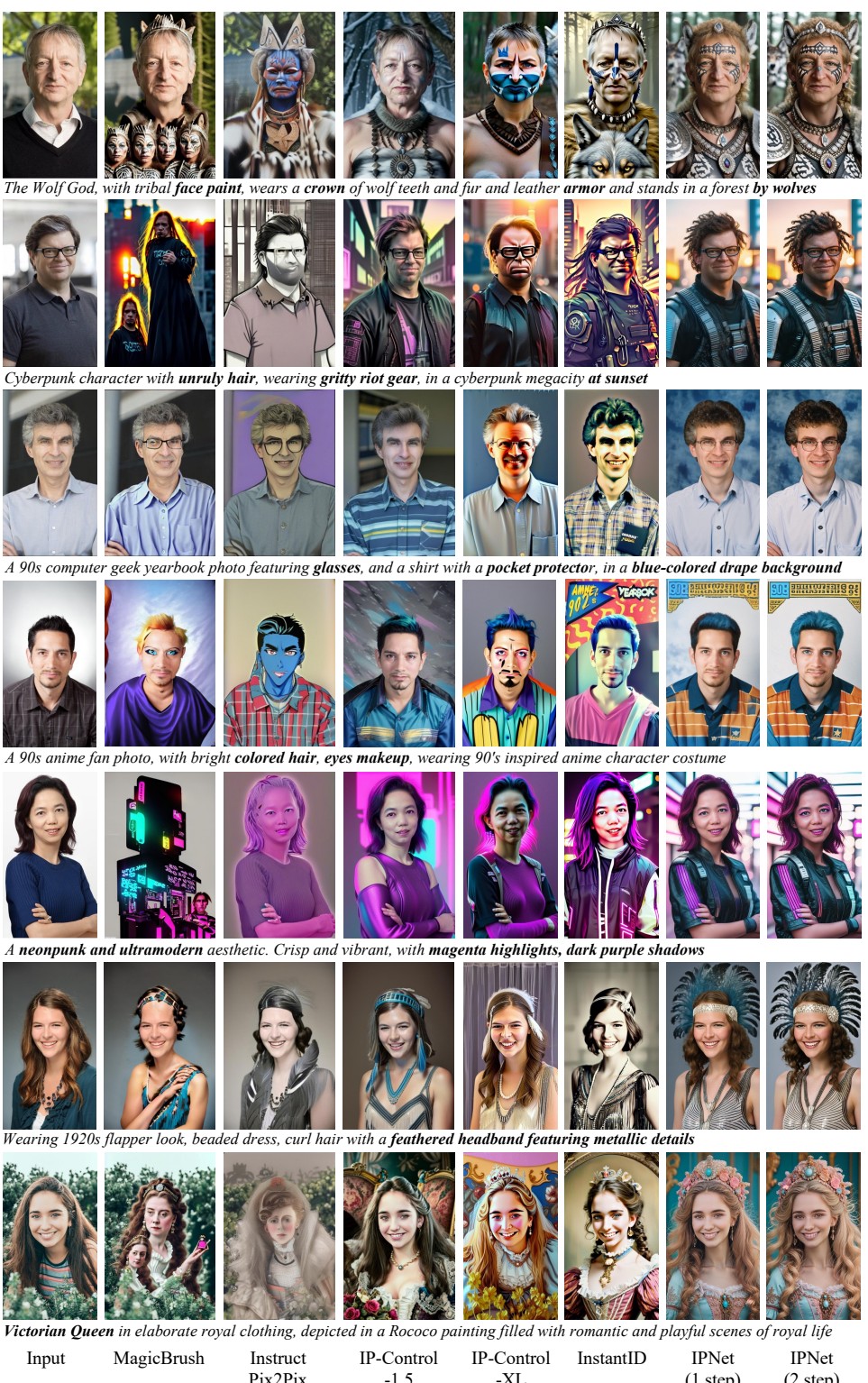

*The Wolf God, with tribal **face paint**, wears a **crown** of wolf teeth and fur and leather **armor** and stands in a forest **by wolves***

*Cyberpunk character with **unruly hair**, wearing **gritty riot gear**, in a cyberpunk megacity **at sunset***

*A 90s computer geek yearbook photo featuring **glasses**, and a shirt with a **pocket protector**, in a **blue-colored drape background***

*A 90s anime fan photo, with bright **colored hair**, **eyes makeup**, wearing 90's inspired anime character costume*

*A **neonpunk and ultramodern** aesthetic. Crisp and vibrant, with **magenta highlights, dark purple shadows***

*Wearing 1920s flapper look, beaded dress, curl hair with a **feathered headband featuring metallic details***

***Victorian Queen** in elaborate royal clothing, depicted in a Rococo painting filled with romantic and playful scenes of royal life*

| Input | MagicBrush | Instruct Pix2Pix | IP-Control -1.5 | IP-Control -XL | InstantID | IPNet (1 step) | IPNet (2 step) |

Figure 5: Qualitative comparison with SOTA methods

Table 2: Quantitative Results of Progressive Improvement over model training and distillation. "+" indicates the inclusion of the specified loss function in the training process.

| Method | Inference Step↓ | Face-Similarity↑ | | Text-Fidelity↑ | | Image-Quality↑ | | |
|---|---|---|---|---|---|---|---|---|
| | | FaceNet | InsightFace | CLIP-Vit-g | CLIP-Vit-H | HPS | Q-Align-Q | Q-Align-A |
| Baseline | 30 | 0.331 | 0.184 | 0.278 | 0.319 | 0.270 | 4.852 | 3.535 |
| $+\mathcal{L}_{\text{cid}}$ | 20 | 0.958 | 0.848 | 0.221 | 0.264 | 0.232 | 4.780 | 3.216 |
| $+\mathcal{L}_{\text{aid}}$ (IDE-Net) | 20 | 0.890 | 0.791 | 0.254 | 0.291 | 0.263 | 4.814 | 3.463 |
| $+\mathcal{L}_{\text{adv}}$ | 1 | 0.048 | 0.029 | 0.197 | 0.236 | 0.189 | 2.436 | 2.167 |
| $+\mathcal{L}_{\text{adv}}+\mathcal{L}_{\text{sds}}$ | 1 | 0.892 | 0.815 | 0.247 | 0.287 | 0.265 | 4.822 | 3.514 |
| $+\mathcal{L}_{\text{adv}}+\mathcal{L}_{\text{distill}}$ | 1 | 0.873 | 0.789 | 0.270 | 0.306 | 0.269 | 4.929 | 3.628 |
| $+\mathcal{L}_{\text{adv}}+\mathcal{L}_{\text{distill}}+\mathcal{L}_{\text{triplet}}$ (IPNet) | 1 | 0.867 | 0.782 | 0.271 | 0.309 | 0.275 | 4.960 | 3.629 |

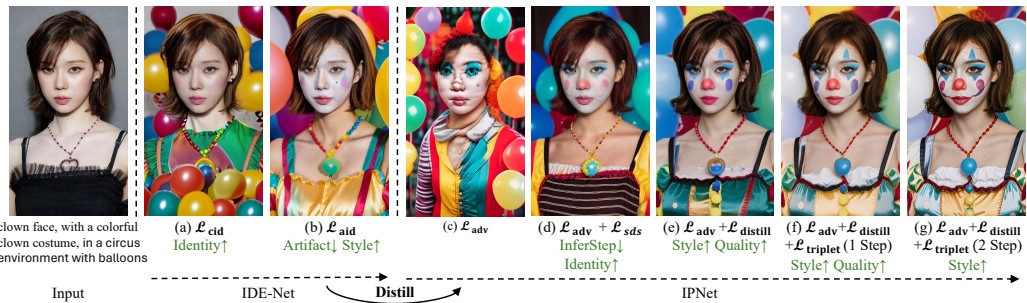

Figure 6: Qualitative results of progressive improvement over model training and distillation

state-of-the-art (SOTA) models. IPNet scores notably high in FaceNet at 0.867, surpassing IP-Control-1.5's 0.794, and leads in InsightFace with 0.782, well ahead of InstantID's 0.689. Additionally, IPNet achieves a CLIP-Vit-g score of 0.271, outperforming InstantID's 0.257, and records a Q-Align-Q score of 4.960, exceeding InstantID's 4.546.

## 4.3 ABLATION STUDY

Table 2 and Figure 6 demonstrate the progressive improvement over model training and distillation. Figure 7 are qualitative comparisons of extra ablation experiments. For a fair comparison across ablation studies, we keep the same training steps and time-step sampling strategy outlined in Section 4.1, while changing the loss function.

### 4.3.1 ANNEALING IDENTITY LOSS

We evaluate IDE-Net using Constant Identity Loss $\mathcal{L}_{\text{cid}}$ and Annealing Identity Loss $\mathcal{L}_{\text{aid}}$. As shown in Table 2, both versions significantly improve Face-Similarity scores compared to the baseline. $\mathcal{L}_{\text{aid}}$ maintains a better balance between Identity-Similarity, Text-Fidelity, and Image-Quality, than $\mathcal{L}_{\text{cid}}$. Furthermore, Figure 6 illustrates that while both $\mathcal{L}_{\text{cid}}$ and $\mathcal{L}_{\text{aid}}$ preserve identity, $\mathcal{L}_{\text{aid}}$ achieve better Text-Fidelity with stronger facial style, consistent with the quantitative results in Table 2. This observation confirms the effectiveness of annealing algorithms in balancing identity preservation and facial style. Further discussion on the selection of annealing algorithms is provided in Appendix E.

### 4.3.2 IDENTITY DISTILLATION LOSS

We validate the effectiveness of Identity Distillation Loss $\mathcal{L}_{\text{distill}}$ for identity preservation through two key observations. First, Table 2 demonstrates that $\mathcal{L}_{\text{adv}}+\mathcal{L}_{\text{distill}}$ significantly enhances FaceNet and InsightFace scores over $\mathcal{L}_{\text{adv}}$, with increases from 0.048 to 0.873 and 0.029 to 0.789, respectively. Second, Figure 6 shows that (e) $\mathcal{L}_{\text{adv}}+\mathcal{L}_{\text{distill}}$ substantially improves identity preservation over (c) $\mathcal{L}_{\text{adv}}$ alone.

We further evaluate DDIM Inversion of $\mathcal{L}_{\text{distill}}$ comparing to the conventional SDS Loss $\mathcal{L}_{\text{sds}}$. First, Table 2 demonstrates that $\mathcal{L}_{\text{distill}}$ surpasses $\mathcal{L}_{\text{sds}}$ in both Text-Fidelity and Image-Quality. Second, in Figure 6, (e) $\mathcal{L}_{\text{distill}}$ output is sharper and more aligned with instruction prompt than (d) $\mathcal{L}_{\text{sds}}$, with minimal difference on identity preservation.

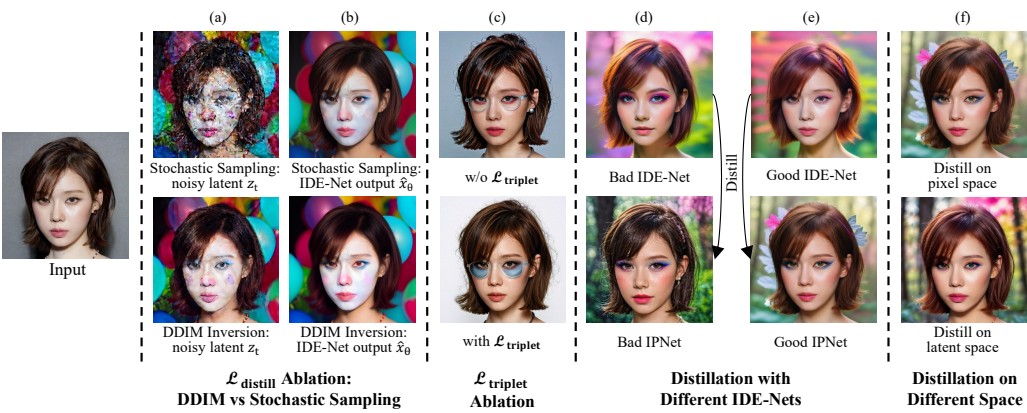

Figure 7: Qualitative comparison of ablation experiments

In Figure 7, we delve deeper into why $\mathcal{L}_{\text{distill}}$ with DDIM Inversion generates superior images over $\mathcal{L}_{\text{sds}}$. Specifically, Figure 7 (a) illustrates that the DDIM Inversion's deterministic sampling generates clearer and more accurate noisy latent codes than $\mathcal{L}_{\text{sds}}$'s stochastic sampling. This clarity leads to improved output quality of IDE-Net, as depicted Figure 7 (b). Consequently, this enhanced guidance from IDE-Net facilitates the distillation of a more effective student model, IPNet.

### 4.3.3 FACE-STYLE ENHANCING TRIPLET LOSS

To evaluate the effect of Face-Style Enhancing Triplet Loss $\mathcal{L}_{\text{triplet}}$ on facial style, we compare models trained with and without $\mathcal{L}_{\text{triplet}}$. According to Table 2, $\mathcal{L}_{\text{triplet}}$ improves Text-Fidelity and Image-Quality and only slightly diminishes Face-Similarity. Figure 7 (c) shows that $\mathcal{L}_{\text{triplet}}$ effectively accentuates instructed facial features, such as glasses, while maintaining identity. The balance between style enhancement and identity preservation is fine-tuned by the margin setting $m$, which varies by dataset. A margin of 0.15 is optimal in our dataset, enhancing style with minimal effect on identity similarity.

### 4.3.4 STYLE BOOST VIA ITERATIVE INFERENCE

We assess the effectiveness of iterative inference by comparing images (f) and (g) in Figure 6.

### 4.3.5 MULTI-OBJECT DISTILLATION

We validate the effectiveness of multi-object distillation through two key observations. First, Table 2 demonstrates that IPNet not only enhances Text-Fidelity and Image-Quality over IDE-Net but also maintains high Face-Similarity. Second, Figure 6 illustrates that images (f) and (g) generated by IPNet have superior facial style over (b) generated by IDE-Net. Additionally, we perform experiments on multi-object distillation using different IDE-Nets. As depicted in Figure 7 (d) and (e), a well-performing IDE-Net teacher model retains high face similarity, significantly boosting the identity preservation capabilities of the student model IPNet during model distillation. This underscores the importance of strong identity preservation in the teacher model for effective distillation. Furthermore, in Figure 7 (f), we observe that distillation in pixel space achieves slightly better facial style over latent space with the same IDE-Net.

## 5 CONCLUSIONS

This study establishes a new benchmark in instruction-based portrait editing, achieving exceptional identity preservation, precise image editing, and rapid model inference. We introduce the Annealing Identity Loss to significantly enhance identity preservation within IDENet and implement the Diffusion Multi-Objective Distillation process to effectively distill IDENet into IPNet. This approach utilizes Adversarial Loss, Identity Distillation Loss, and Face-Style Enhancing Triplet Loss to address multiple critical objectives simultaneously.

ETHICS STATEMENT

Our approach enhances the quality of portrait editing and enables users to create facial portraits in the styles they prefer. We encourage the responsible use of this technology, avoiding the generation of output that could be deemed inappropriate or harmful.

REPRODUCIBILITY STATEMENT

We have made significant efforts to ensure the reproducibility of the results presented in this paper. Detailed instructions for dataset generation are provided in Appendix B. Model training procedures are outlined in Section 3, while the main experiments and evaluation are discussed in Section 4. Additional experimental results can be found in Appendix E. Moreover, a demo video is also available for download in the Supplementary Materials. We hope these resources will facilitate the replication of our findings and encourage further research building on our work.

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

SUPPLEMENTARY MATERIAL

## A RELATED WORK

### A.1 TEXT-BASED IMAGE EDITING

Recent advancements in image editing leverage pre-trained text-to-image diffusion models. The Prompt-to-Prompt (P2P) Hertz et al. (2022) utilizes cross-attention maps to enable targeted edits based on text prompts. In contrast, SmartBrush Xie et al. (2023) enhances inpainting by integrating text and shape guidance. SDEdit Meng et al. (2021) applies noise followed by denoising via a stochastic differential equation, without requiring specialized training. Further, large datasets have been generated to refine user interfaces and facilitate instruction-based editing. For instance, InstructPix2Pix Brooks et al. (2023) uses a synthetic dataset to train models for guided image editing. Similarly, MagicBrush Zhang et al. (2024) and Emu Edit Sheynin et al. (2023) develop datasets and employ multi-task training to expand the capabilities and intuitiveness of image editing tools. Existing methods prioritize instruction accuracy but neglect the specific needs of portrait editing, such as identity preservation. Our approach uniquely focuses on portrait image editing, ensuring identity integrity while accurately following instructions.

### A.2 IDENTITY-PRESERVING IMAGE GENERATION

ID-preserving image generation techniques, such as IGAR Piao et al. (2021) and CGOF series Sun et al. (2022; 2023), focus on retaining key facial attributes essential for identity recognition in practical applications. PhotoMaker Li et al. (2023b) enhances these features by adjusting Transformer layers and integrating class and image embeddings. IP-Adapter Ye et al. (2023) introduces a specialized cross-attention mechanism, leveraging reference images as visual prompts to improve text-visual data integration. InstantID Wang et al. (2024a) incorporates IdentityNet to preserve detailed attributes from reference portraits. FastComposer Xiao et al. (2023) employs localized cross-attention to prevent identity blending issues common in text-to-image models, ensuring more accurate feature representation. All methods share a common issue: adding identity weakens the image's style due to inadequate exploration of balancing style and identity. Moreover, using only identity embeddings as input without targeted supervision fails to effectively preserve identity.

### A.3 STABLE DIFFUSION MODEL DISTILLATION

Diffusion models' iterative denoising steps hinder real-time application Li et al. (2024); Wu et al. (2024). To enhance their speed, approaches such as DPM-Solver Lu et al. (2022), DDIM Song et al. (2020), and DEIS Zhang & Chen (2022) have been developed to accelerate the sampling process. Other advancements include Progressive Distillation Salimans & Ho (2022) and Guided Distillation Meng et al. (2023), which reduce the number of sampling steps from thousands to as few as 4-8. Additionally, recent innovations in model distillation, such as Score Distillation have extended capabilities to 3D synthesis Poole et al. (2022); Wang et al. (2023). Moreover, the adversarial diffusion models Sauer et al. (2023b); Xiao et al. (2021) further enhance performance by integrating GANs and adversarial training, marking significant strides in diffusion model efficiency and application scope. Although distillation methods enhance the speed of diffusion models, they inevitably lead to a reduction in performance.

## B DATASET

Due to a lack of publicly available datasets that adequately preserve identity for portrait image editing, we create a specialized dataset. Each entry $(c_I, c_T, x)$ in this dataset includes an input image $c_I$, an instruction prompt $c_T$, and a target image $x$. Figure 8 outlines the framework for dataset generation, which includes prompt generation, image pairs generation, and image post-processing. Table 3 includes the basic information of the dataset, including the size, image resolution, and face similarity score.

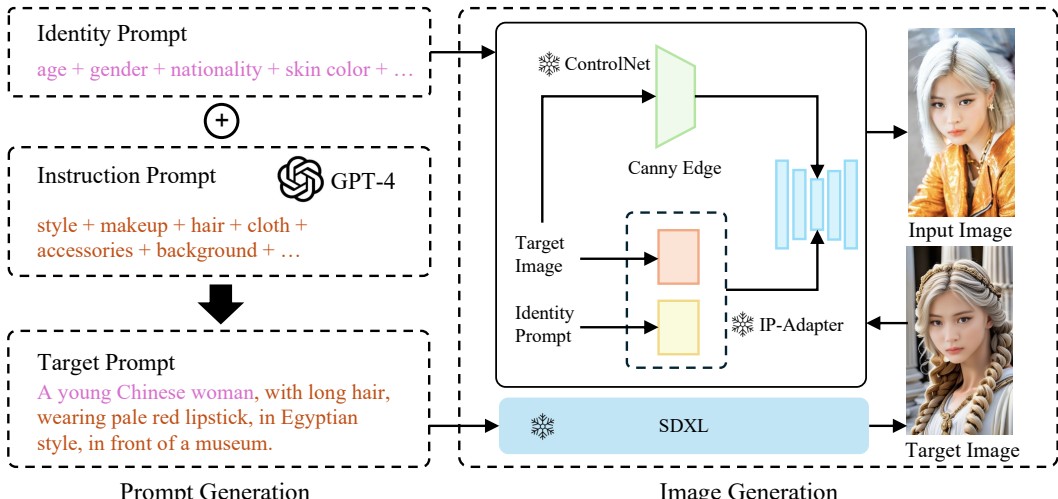

Figure 8: Dataset Generation Pipeline

Table 3: Basic Information of the synthetic training dataset

| | Data Pair Number | Image Resolution | Face-Similarity↑ | |
| --- | --- | --- | --- | --- |
| | | | FaceNet | InsightFace |
| Dataset | 10,000,000 | 1024x576 | 0.749 | 0.576 |

### B.1 PROMPT GENERATION

From Figure 8, we design **identity prompts** to define human attributes such as age, gender, nationality, and skin color for generating input images, resulting in 10 million unique prompts. **Instruction prompts**, crafted using GPT-4, direct the transformation of these input images into various styles, makeups, hairstyles, and backgrounds. We generate in total of 10 million instruction prompts. We then combine these identity and instruction prompts and sample 10 million target prompts, which are used for generating target images.

### B.2 IMAGE PAIR GENERATION

We deviate from traditional methods that first generate the input image and then create the target image using instruction prompts. Our approach starts by creating the target image using the SDXL model without control inputs or adapters, ensuring the generated image closely aligns with the instruction prompt, thereby improving text alignment in our model. The input image is subsequently produced under controlled conditions based on the target image, as illustrated in Figure 8. To improve identity and pose consistency between the images, we utilize IP-Adapter Ye et al. (2023), fine-tuned on the target images, and ControlNet Zhang et al. (2023), leveraging canny edge. However, identity preservation between the input image and the target image is still insufficient. This issue is effectively addressed by introducing a novel identity loss in Section 3.1.1. Using the prompt sets described in Section B.1, we generate 10 million image pairs within this framework.

## C QUALITATIVE COMPARISON BETWEEN IDE-NET AND IPNET

As illustrated in Figure 9, the first row presents the outputs of IDE-Net, while the second row showcases the outputs of IPNet. While IDE-Net demonstrates slightly superior identity preservation, IPNet excels in overall image quality and style attributes, such as makeup, glasses, and masks. These qualitative observations are consistent with the quantitative results reported in Table 2.

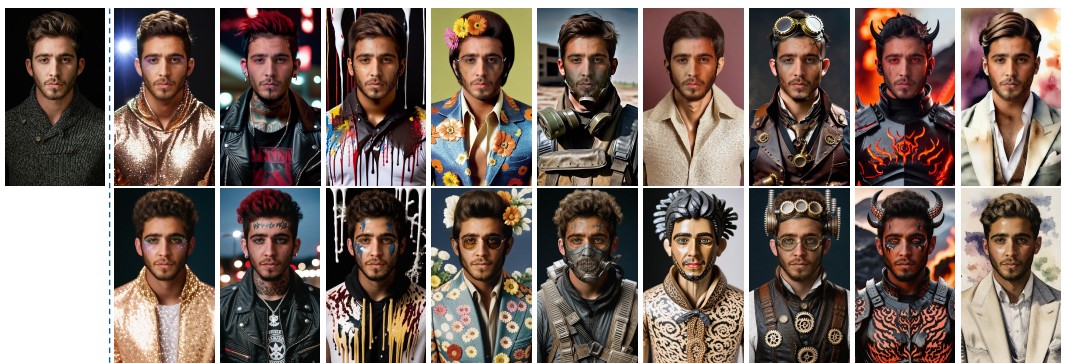

Input        The first row is the output of IDE-Net and the second row is the output of IPNet

Figure 9: Qualitative Comparison between IDE-Net and IPNet

## D  Image Diversity Comparison between IDE-Net and IPNet

Intra-LPIPS (Intra-Learned Perceptual Image Patch Similarity) is a metric for evaluating the diversity of generated images Zhu et al. (2022). To compute Intra-LPIPS for a model, we generate multiple images from a single portrait input using different instruction prompts and calculate the pairwise LPIPS among the generated images. For multiple portrait inputs, the mean Intra-LPIPS is computed by averaging the Intra-LPIPS values across all inputs. The mean Intra-LPIPS scores are shown in Table 4.

The mean Intra-LPIPS score of the student model IPNet is higher than the teacher model IDE-Net, because we applied the proposed loss functions during the distillation process. According to Table 2, IPNet outperforms IDE-Net in text fidelity (including facial style) and image quality, although it shows a slight regression in face similarity. The improved style fidelity but weaker identity preservation indicates that, for the same input image and different text prompts, the variations among the images generated by the IPNet are greater than those produced by the IDE-Net. As a result, the student model IPNet achieves a higher Intra-LPIPS score.

Table 4: Intra-LPIPS of IDE-Net and IPNet

| Metrics | IDE-Net | IPNet |
|---|---|---|
| Intra-LPIPS | 0.571 | 0.589 |

## E  Annealing Algorithm Selection For Annealing Identity Loss

We tried linear annealing $\frac{a*t}{T_{max}}+b$ and cosine annealing $0.5*\alpha*(1+cosine(pi*\frac{t}{T_{max}}))+(1-\alpha)$ algorithms for Annealing Identity Loss. As demonstrated in Table 5, linear annealing algorithms outperform cosine annealing in Text-Fidelity and Image-Quality but show lower Face-Similarity. Given the already high scores in Face-Similarity, which render the visual distinction between the output and input identities negligible, we prioritize superior text fidelity and image quality by adopting $\frac{t}{T_{max}}$.

Table 5: Quantitative results of the ablation experiment for Annealing Identity Loss. The best results are in **bold**.

| Annealing | Face-Similarity↑ | | Text-Fidelity↑ | | Image-Quality↑ | | |
|---|---|---|---|---|---|---|---|
| | FaceNet | InsightFace | CLIP-Vit-g | CLIP-Vit-H | HPS | Q-Align-Q | Q-Align-A |
| $\frac{t}{T_{max}}$ | 0.890 | 0.791 | **0.254** | **0.291** | **0.263** | **4.814** | **3.463** |
| $\frac{t}{T_{max}}+0.1$ | 0.903 | 0.802 | 0.234 | 0.272 | 0.255 | 4.781 | 3.327 |
| $0.5*(1+cosine(pi*\frac{t}{T_{max}}))$ | 0.895 | 0.799 | 0.233 | 0.273 | 0.261 | 4.783 | 3.422 |
| $0.45*(1+cosine(pi*\frac{t}{T_{max}}))+0.1$ | **0.911** | **0.809** | 0.226 | 0.269 | 0.249 | 4.781 | 3.309 |

Table 6: Training Parameters

| Model | Time step | Loss function | Training batch size | Training steps |
|---|---|---|---|---|
| IDE-Net | [0, 999] | $\mathcal{L}_{\text{dm}} + 0.7 \cdot \mathcal{L}_{\text{aid}}$ | 256 | 40k |
| IPNet (High time step) | [400, 800] | $\mathcal{L}_{\text{adv}} + 1 \cdot \mathcal{L}_{\text{distill}} + 0 \cdot \mathcal{L}_{\text{triplet}}$ | 256 | 25k |
| IPNet (Middle time step) | [200, 400] | $\mathcal{L}_{\text{adv}} + 0.3 \cdot \mathcal{L}_{\text{distill}} + 0 \cdot \mathcal{L}_{\text{triplet}}$ | 256 | 15k |
| IPNet (Low time step) | [150, 200] | $\mathcal{L}_{\text{adv}} + 0.3 \cdot \mathcal{L}_{\text{distill}} + 1 \cdot \mathcal{L}_{\text{triplet}}$ | 2048 | 1k |

## F  Necessity of Two-Stage Training

Directly incorporating the Annealing Identity Loss (Section 3.1.1), Adversarial Loss (Section 3.2.1), Identity Distillation Loss (Section 3.2.2), and Face-Style Enhancing Triplet Loss (Section 3.2.3) in one stage would not allow it to match IPNet's performance due to the following reasons:

First, model distillation is necessary for one-step inference. Without distillation from a teacher model, training a one-step inference diffusion model IPNet with adversarial loss is similar to training a GAN. While GAN is faster than the diffusion model, GAN suffers from issues like mode collapse and reduced diversity. This is because GANs rely on single-step inference, whereas diffusion models involve multiple steps of noise addition and denoising, which naturally enhance diversity. Dhariwal & Nichol (2021) By using a diffusion model as a teacher to distill a GAN-like model with distillation loss, we can combine the strengths of both approaches: achieving the good diversity of diffusion models and the fast, one-step inference capability of GAN-like models. Wang et al. (2022)

Second, two-stage training is crucial for balancing identity preservation and facial style. Training diffusion model with all losses combined in a single stage does not achieve a proper balance between identity preservation and facial style. We experimented extensively with various loss weight combinations during single-stage training and observed that the model failed to converge effectively. Specifically, the Annealing Identity Loss and the Face-Style Enhancing Triplet Loss oscillated during training. This instability arises because identity preservation and facial style enhancement are inherently conflicting tasks, making it difficult for a model to learn both effectively in a single stage.

## G  Training Details

The model training parameters are provided in the Table 6, including the denoising sampling step, loss weights, training batch size, and training step.

### G.1  IDE-Net Training

For the teacher model training, the time steps are sampled within the range [0, 999]. We selected 0.7 as Annealing Identity Loss weight to achieve a good balance between identity preservation and image quality. In addition, we compared different annealing algorithms with different weights in Appendix E.

### G.2  IPNet Training

The training of the student model IPNet can be counted as three stages based on the sampled time steps:

**High Time Step** In this stage, the teacher model samples time steps within the range [400, 800]. The loss function in this stage combines Adversarial Loss and Identity Distillation Loss (stochastic noise sampling). The primary goal is to distill the overall structure and pose alignment. The weight of Identity Distillation Loss is set to 1 because the student model is just starting training, the discriminator is weak giving noisy feedback, and thus requires stronger supervision from the teacher. If the weight is set to too small, like 0, the training of the student model is likely to collapse at an early stage, as shown in Figure 6 image (c).

**Middle Time Step** The time steps in this stage are sampled within the range of [200, 400]. The denoising target of the teacher model in the high sampling step introduces the artifacts, which can be removed in the lower sampling time step in this stage. The pose alignment is further strengthened in this stage. Therefore, the time-step threshold is determined to be 200 by checking the accuracy of spatial alignment and image quality during training. The Identity Distillation Loss (stochastic noise sampling) weight is reduced to 0.3, to weigh more on adversarial loss and improve the image quality of few-step generation.

**Low Time Step** In this stage, the time step is sampled within the range of [150, 200] to refine details and enhance style. The loss function is expanded to include: Adversarial Loss, Identity Distillation Loss (DDIM inversion), and Face-Style Enhancing Triplet Loss. Building upon the solid pose alignment for the student model achieved in the high-time-step and middle-time-step stages, this stage is mainly for refining details and enhancing image quality and style: First, the Identity Distillation Loss (DDIM inversion) allows fine-grained distillation, which improves the text fidelity and image quality, compared to Identity Distillation Loss (stochastic noise sampling) in the previous stages; Second, the Face-Style Enhancing Triplet Loss added in this stage strengthens style consistency while preserving identity. A large batch size of 2048 is used to incorporate a greater variety of facial styles within each batch, to stabilize training with the Face-Style Enhancing Triplet Loss. Therefore, the training steps of IPNet are reduced accordingly due to the large batch size.

## H    LIMITATION OF FACIAL EXPRESSION EDITING

Our approach effectively edits static features like makeup, accessories, style, clothes, and background. However, dynamic attributes like facial expressions are less explored. Facial expression editing was not the primary focus during dataset creation, resulting in a limited number of expression-related image pairs in our dataset. Consequently, our method performs less effectively on expression editing (as shown in Figure 10) compared to other features including background, clothing, jewelry, hair, and makeup edits. However, it still surpasses state-of-the-art methods in this domain, as demonstrated in Figure 11 with the examples of prompts including "smiling".

To address this limitation, we plan to expand our dataset in future iterations by adding more diverse facial expression image pairs, such as "crying", "laughing", and "angry". This will significantly enhance the model's ability to handle facial expression editing. This improvement is theoretically feasible, as the identity preservation loss, Annealing Identity Loss (introduced in Section 3.1.1) is calculated at the embedding level rather than the pixel level, ensuring identity preservation while allowing adjustments to facial expressions.

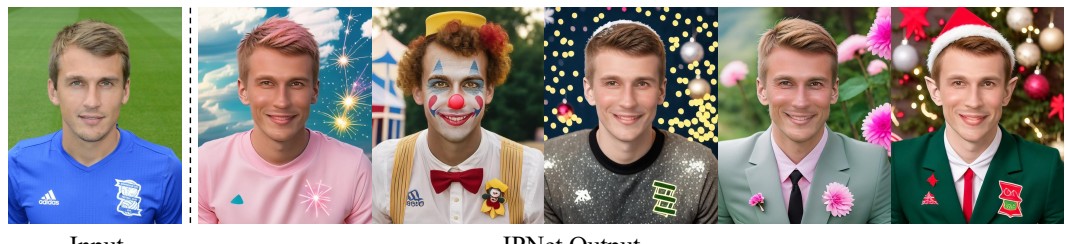

Input                                    IPNet Output

Figure 10: The output of IPNet with the instruction prompts including "smiling"

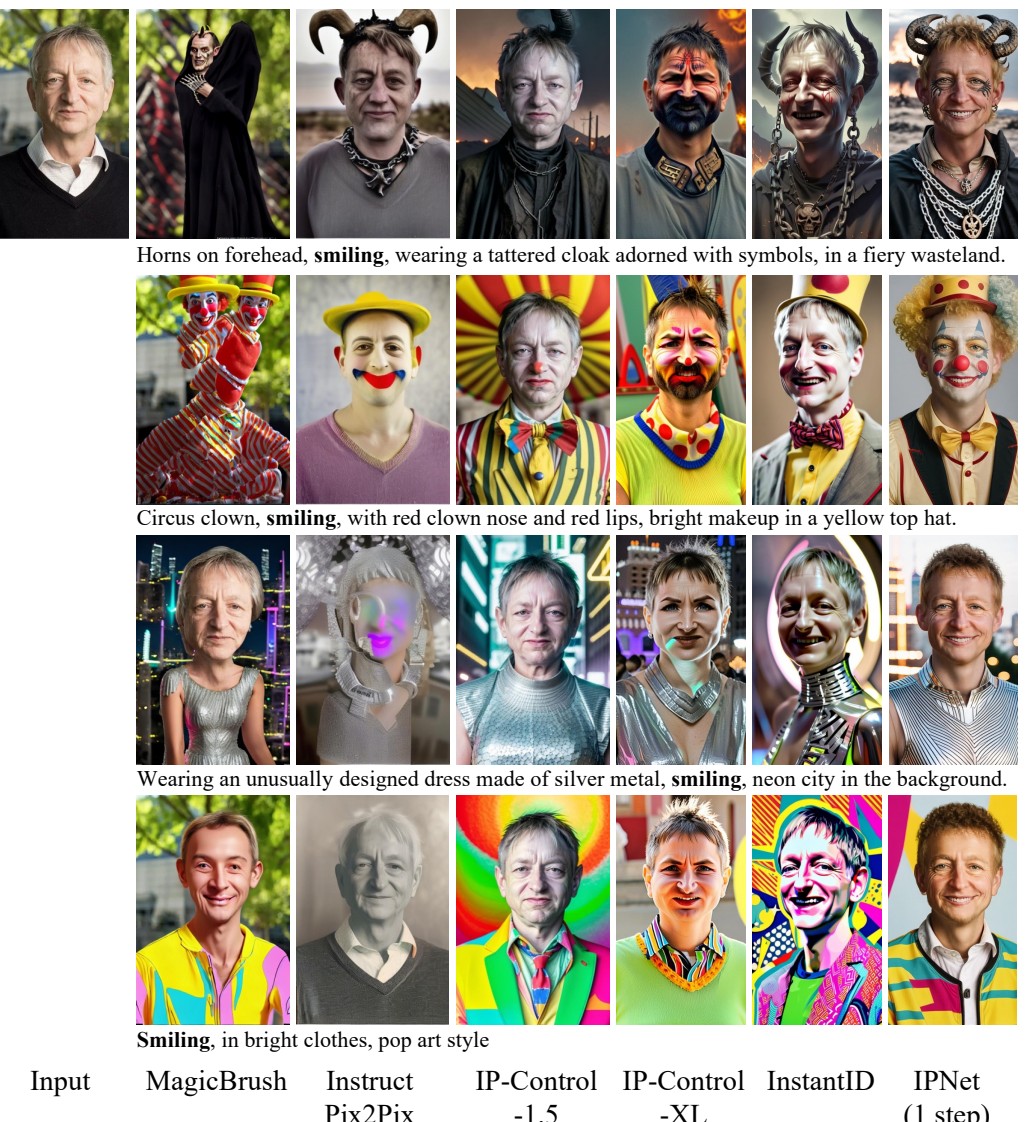

Horns on forehead, **smiling**, wearing a tattered cloak adorned with symbols, in a fiery wasteland.

Circus clown, **smiling**, with red clown nose and red lips, bright makeup in a yellow top hat.

Wearing an unusually designed dress made of silver metal, **smiling**, neon city in the background.

**Smiling**, in bright clothes, pop art style

| Input | MagicBrush | Instruct Pix2Pix | IP-Control -1.5 | IP-Control -XL | InstantID | IPNet (1 step) |

Figure 11: Qualitative comparison with SOTA methods for the prompts including "smiling"

# I   PROMPT EXAMPLES

**The prompts used to generated images in Figure 1 from left to right are below**:

In the first row:

(1) Adorable figure crafted from paper mache

(2) A playful doodle with uneven lines, unusual proportions, and a basic grasp of perspective, drawn with crayons on rough paper

(3) A fusion of organic and mechanical elements in a biomechanical style, highly detailed and intricate, with a futuristic and cybernetic aesthetic

(4) Greek frescoes, showcasing ancient mythology with a harmonious composition and meticulous detailing

(5) A graphic poster featuring a beautiful face with plump lips and orange sunglasses, surrounded by multicolored flowers and vibrant hues, radiating a retro 70's vibe

(6) A joyful clown with a cheerful face, dressed in a vibrant, colorful costume, set in a lively circus environment with bold color contrasts and the circus in the background

(7) Realistic Gothic fantasy scene with a necromancer in ragged, bone-decorated garments, surrounded by a graveyard and animated skeletons

(8) A demon with intense red eyes and sharp, angular features, dressed in armor marked by glowing red runes, standing in a blazing infernal landscape with molten lava, smoky haze, and glowing ember effects

In the second row:

(1) A watercolor painting in the style of the 1920s Gatsby era

(2) Valentine's Day card inspired by 1950s elegance, showcasing a pink silk shirt, a heart and flower-filled background, and a blend of artistic modern aesthetics

(3) Lowbrow art featuring a 2D rainbow, infused with bright, surreal elements and modern surrealism, bursting with rainbow colors

(4) Pharaoh of Ancient Egypt, wearing classic Egyptian makeup, embodying the culture and style of the ancient civilization

(5) Wearing lip gloss, a pink strapless tank top, a denim skirt, and a pink handbag slung over the shoulder, with pink sunglasses and a street scene in the background

(6) An alchemist dressed in a hooded robe adorned with mystical symbols, set in a mysterious stone laboratory filled with vials, bubbling potions, flasks, and glowing gemstones

(7) Wearing luxurious Venetian masquerade mask with intricate designs, in an ornate palace garden with fountains

(8) Wearing rave makeup and a vibrant rave outfit, with a laser show lighting up the background at the beach

## J    MORE GENERATED IMAGES BY IPNET

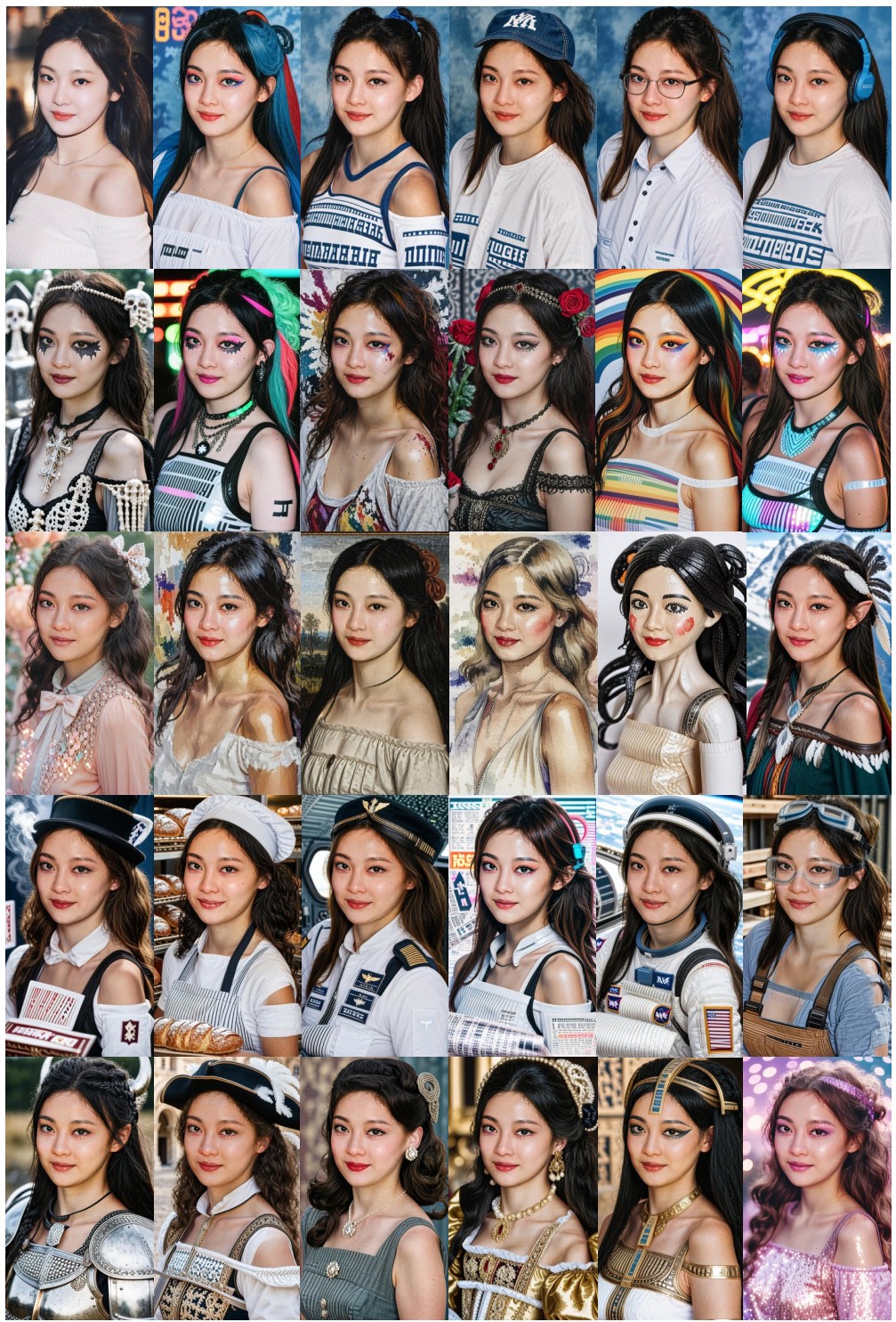

Figure 12: More Generated Images by IPNet

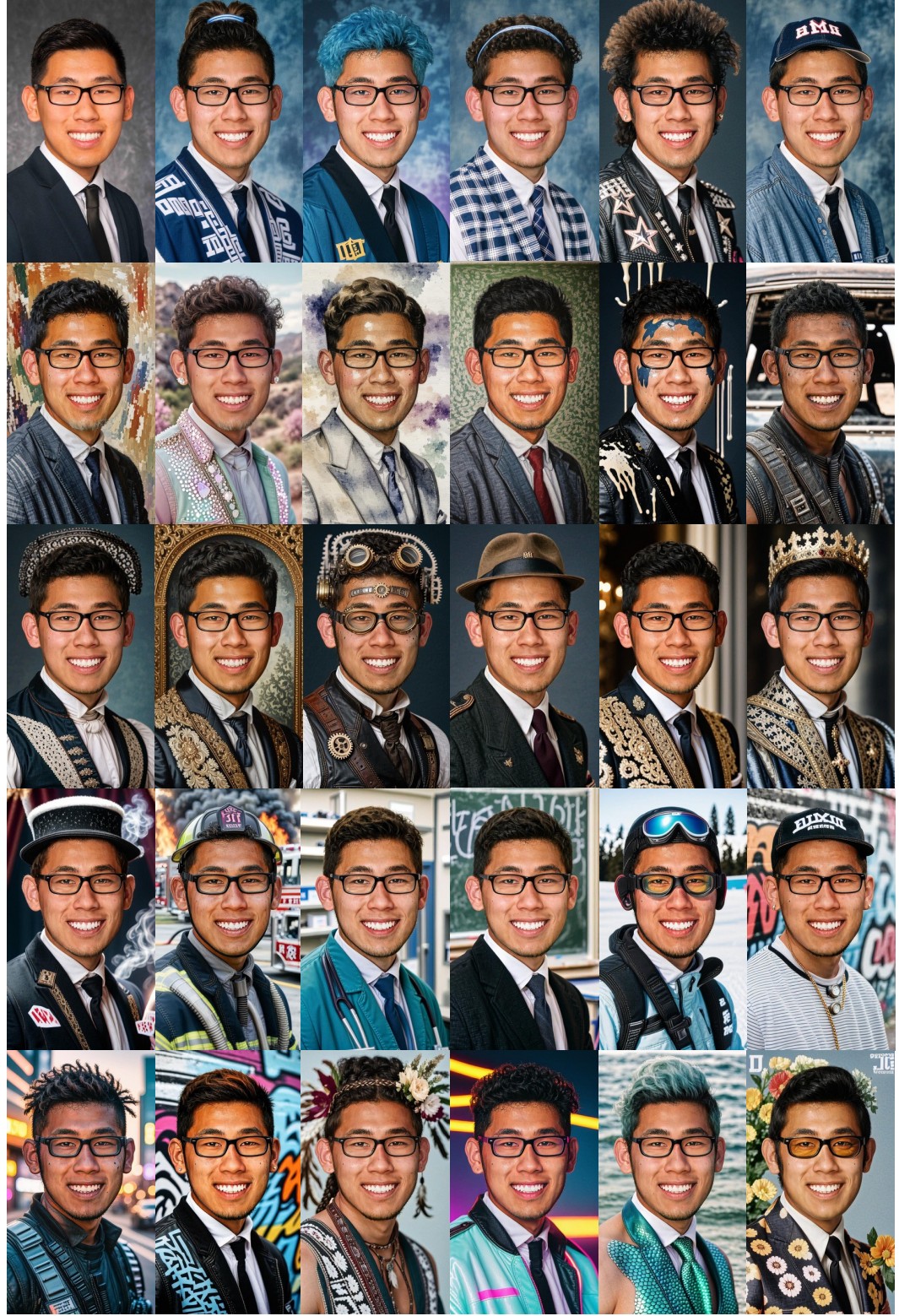

Figure 13: More Generated Images by IPNet

## K    Resources and Pre-trained Models

We provide the details of the resources and pre-trained models used in this work, along with their respective URLs in Table 7.

| Model/Resource | Details and URL |
|---|---|
| **IP-Control-XL** | *IPAdapter (ip-adapter-faceid_sdxl.bin):* `https://huggingface.co/h94/IP-Adapter-FaceID` |
| | *ControlNet-xl (canny):* `https://huggingface.co/diffusers/controlnet-canny-sdxl-1.0` |
| **IP-Control-1.5** | *IPAdapter (ip-adapter-faceid-plus_sd15.bin):* `https://huggingface.co/h94/IP-Adapter-FaceID` |
| | *ControlNet-1.5 (control_sd15_canny.pth):* `https://huggingface.co/lllyasviel/ControlNet` |
| **Instruct-Pix2Pix** | *diffusers/sdxl-instructpix2pix-768:* `https://huggingface.co/diffusers/sdxl-instructpix2pix-768` |
| **MagicBrush** | `https://huggingface.co/osunlp/InstructPix2Pix-MagicBrush` |
| **InstantID** | `https://huggingface.co/InstantX/InstantID` |

Table 7: Resources and Pre-trained Models with URLs.

