# OpenReview forum: "InstantPortrait: One-Step Portrait Editing via Diffusion Multi-Objective Distillation"
_ICLR.cc/2025/Conference — ICLR 2025 Poster_

### Official Review · Reviewer_B5oG · 2024-10-27

**Soundness:** 3
**Presentation:** 2
**Contribution:** 3
**Rating:** 6
**Confidence:** 4

**Summary:**

This paper proposes a one-step text-based portrait image editing method, which can achieve the balance between identity preservation and text alignment. They firstly train an Identity Enhancement Network (IDE-Net) to ensure robust identity preservation. Then, they propose a diffusion Multi-Objective Distillation process to distill IDENet into IPNet, achieving one-step instruction-based portrait image editing.

**Strengths:**

1. They propose to distill IDENet into IPNet with the diffusion Multi-Objective Distillation process, achieving one-step instruction-based portrait image editing.
2. They propose an Annealing Identity Loss that balances identity preservation and text alignment.
3. They design a dataset generation pipeline to generate a large-scale paired dataset, where each pair contains: 1) an input image; 2) a text prompt; and 3) a target image generated by SDXL with IPAdapter and ControlNet.
4. Experiment results show that they can generate high-quality image editing results with better identity preservation.

**Weaknesses:**

1. This paper primarily allows editing only the style of portraits; it is not suitable for modifying expressions, poses, or the subject’s position within the image, which imposes certain limitations.
2. The necessity of IDENet remains unclear. Can you provide a more detailed justification for training IDENet rather than using existing methods like IPAdapter or InstantID, and discuss how this choice impacts the overall performance of IPNet?

**Questions:**

1. Is IDE-Net purely a reconstruction model? Does it take text as input, and could you provide some sample results?
2. If we directly train IDE-Net with the adversarial loss from 3.2.1, the triplet loss from 3.2.3, and the original IDENet losses, would the results surpass those of IPNet?
3. When performing distillation, since IPNet only involves a single step of sampling, how is the number of noise-adding and denoising steps for IDE-Net determined?
4. Given that the proposed dataset is paired, why wasn’t diffusion loss, some photometric loss, or the losses used in IDENet, applied when training IPNet? Can using those losses provide a more accurate supervision signal than IDENet’s distillation loss?


I look forward to the author’s response, and I am willing to raise my score if some of my concerns are addressed.

---

> ### Author Response · Authors · 2024-11-20
> **Response to Reviewer B5oG [Part 1/5]**
>
> We sincerely thank the reviewer for your insightful comments, valuable feedback, and recognition of our work, particularly for acknowledging:
>
> - Diffusion-based Multi-Objective Distillation enables distilling IDENet into IPNet for one-step instruction-based portrait editing.
> - The Annealing Identity Loss to balance identity preservation and text alignment.
> - Our dataset generation pipeline enables large-scale paired data creation.
> - Our experimental results demonstrate high-quality editing with improved identity preservation.
>
> Based on your feedback and suggestions, we have carefully revised the paper. The following updates and enhancements have been implemented accordingly:
>
> - We added more details, analysis, and plans about our limitations in the Supplementary Material Section “Limitation of Facial Expression Editing”.
> - We compared the facial expression editing with SOTA models and added it in the Supplementary Material Section “Limitation of Facial Expression Editing”.
> - We added a clear statement at the beginning of Section 3.1, highlighting that IDE-Net uses text as part of its input.
> - We provided the text prompt for Figure 6 to clarify that both IDE-Net and IPNet rely on text and image inputs.
> - We added a new section “Qualitative Comparison between IDE-Net and IPNet” and provided more examples of IPNet in Figure 9 in the supplementary material.
> - We added a new section titled “Necessity of Two-Stage Training” in the Supplementary Material.
> - We added a new section titled “Training Parameter” in the Supplementary Material.
>
> Once again, thank you for your detailed and constructive input, which has been instrumental in improving the overall quality of our work. Below, we present detailed responses to each of your questions:
>
> > **Q1: This paper primarily allows editing only the style of portraits; it is not suitable for modifying expressions, poses, or the subject’s position within the image, which imposes certain limitations.**
>
> **A1:** Thanks for your comments! Our work focuses on real-time, instruction-based portrait editing, which is particularly valuable for applications such as augmented reality (AR), video communication, and filter effects. In such scenarios, preserving the subject's original pose and position is often essential to meet the practical requirements of real-world use cases.
>
> Except for the position, our work supports a wide range of portrait-related edits, including changes to makeup, accessories, style, clothes, hair, background, facial expressions, and combinations of these attributes.
> - For makeup, accessories, style, clothes, hair, and background attributes, our dataset was primarily focused on, thus resulting in a strong performance for these features.
> - Facial expression editing was not the primary focus during dataset creation, resulting in a limited number of expression-related image pairs in our dataset. Consequently, our method performs less effectively on expression editing (as shown in paper Figure 10) compared to other attributes, such as background and facial style. However, it still surpasses state-of-the-art methods in this domain, as demonstrated in paper Figure 11 with the examples of prompts including "smiling". To address this limitation, we plan to expand our dataset in future iterations by adding more diverse facial expression image pairs, such as "crying", "laughing", and “angry”. This will significantly enhance the model’s ability to handle facial expression editing. This improvement is theoretically feasible, as the identity preservation loss, Annealing Identity Loss (in paper section 3.1.1) is calculated at the embedding level rather than the pixel level, ensuring identity preservation while allowing adjustments to facial expressions.
>
>
> We hope this explanation clarifies the scope and potential of our method, and we are committed to addressing these limitations in our future work. To avoid confusion per your suggestion, we added more details, analysis, and plans about our limitations in the Supplementary Material Section “Limitation of Facial Expression Editing”. In addition, we compared the facial expression editing with SOTA models and added it to the Supplementary Material Section “Limitation of Facial Expression Editing”.  Please let us know if you have further questions or concerns.

---

> ### Author Response · Authors · 2024-11-20
> **Response to Reviewer B5oG [Part 2/5]**
>
> > **Q2: The necessity of IDENet remains unclear. Can you provide a more detailed justification for training IDENet rather than using existing methods like IPAdapter or InstantID, and discuss how this choice impacts the overall performance of IPNet?**
>
> **A2:** Thank you for this insightful question! Detailed justifications are below:
>
> **1)** Why is IDE-Net necessary? Why not use existing methods like IPAdapter or InstantID?
>
> We introduce IDE-Net because it enhances identity consistency significantly. IPAdapter relies solely on cross-attention for feature extraction and lacks strong identity supervision [1]. In contrast, IDE-Net uses Annealing Identity Loss to explicitly preserve input image identity.
>
> Our experiment results demonstrate that existing methods, such as IPAdapter and InstantID, fail to meet this standard while IDENet can. As shown below, the identity preservation of IPAdapter and InstantID are much worse than IDE-Net's scores.
>
> | **Model**      | **FaceNet Score ↑** | **InsightFace Score ↑** |
> |-----------------|-------------------|------------------------|
> | **IPAdapter**   | 0.794             | 0.670                 |
> | **InstantID**   | 0.751             | 0.689                 |
> | **IDE-Net**     | 0.890             | 0.791                 |
>
>
> **2)** How does choosing IDE-Net over IPAdapter or InstantID impact IPNet’s overall performance?
>
> IDENet serves as the teacher model for IPNet, directly influencing IPNet’s ability to preserve identity. As explained in paper Section 4.3.5 Multi-Object Distillation, we performed an ablation study on multi-object distillation using different IDE-Nets. **As depicted in the paper Figure 7 (d) and (e), a well-performing IDE-Net teacher model retains high face similarity, significantly boosting the identity preservation capabilities of the student model IPNet during model distillation.** This underscores the importance of strong identity preservation in the teacher model for effective distillation.
>
> Therefore, if we choose a teacher model with a worse face similarity score like IPAdapter or InstantID, the student model IPNet will be worse at identity preservation.
>
> ******
>
> > **Q3: Is IDE-Net purely a reconstruction model? Does it take text as input, and could you provide some sample results?**
>
> **A3:** Thanks for your questions!
>
> **1)** IDE-Net is not purely a reconstruction model, but a generative model, with superb identity preservation. IDE-Net takes both text and images as input. In section 4.3 Ablation Study, images (a) and (b) in Figure 6 were produced by IDE-Net with the textual prompt of “clown face, with a colorful clown costume, in a circus environment with balloons”. These images preserve the identity of the input image while incorporating edits to the clothing and background based on the prompt. Although images (a) and (b) display minimal clown makeup, this limitation was addressed during IP-Net training through the application of Multi-Objective Distillation, as demonstrated in images (f) and (g).
>
>
> **2)** We added a new section “Qualitative Comparison between IDE-Net and IPNet” in the supplementary material and provided more examples of IPNet in Figure 9. From the Figure 9, while IDE-Net demonstrates slightly superior identity preservation, IPNet excels in overall image quality and style attributes, including makeup, glasses, and masks. These qualitative observations are consistent with the quantitative results reported in the main paper Table 2.
>
> Thanks for the great question! To improve clarity and address potential confusion, we have made the following updates:
> - We added a clear statement at the beginning of Section 3.1, highlighting that IDE-Net uses text as part of its input.
> - We provided the text prompt for Figure 6 to clarify that both IDE-Net and IPNet rely on text and image inputs.
> - We added a new section “Qualitative Comparison between IDE-Net and IPNet” and provided more examples of IPNet in Figure 9 in the supplementary material.

---

> ### Author Response · Authors · 2024-11-20
> **Response to Reviewer B5oG [Part 3/5]**
>
> > **Q4: If we directly train IDE-Net with the adversarial loss from 3.2.1, the triplet loss from 3.2.3, and the original IDENet losses, would the results surpass those of IPNet?**
>
> **A4:** Thank you for your insightful question! IPNet demonstrates superior performance over IDE-Net in two key aspects:
> - **achieving a better balance between identity preservation, quality, and facial style**
> - **enabling one-step inference**.
>
> Directly incorporating the adversarial loss and triplet loss into IDE-Net training in one stage training would not allow IDE-Net to match IPNet's performance in either aspect due to the following reasons:
>
> - **Two-stage training is crucial for balancing identity preservation and style:**
> Training IDE-Net with all losses combined in a single stage does not achieve a proper balance between identity preservation and facial style. We experimented extensively with various loss weight combinations during single-stage training and observed that the model failed to converge effectively. Specifically, the Annealing Identity Loss (original IDE-Net loss) and the Face-Style Enhancing Triplet Loss oscillated during training. This instability arises because identity preservation and facial style enhancement are inherently conflicting tasks, making it difficult for a model to learn both effectively in a single stage.
>
> - **Model distillation / two-stage is necessary for one-step inference:**
> Without distillation from a teacher model, training a one-step inference diffusion model IPNet with adversarial loss is similar to training a GAN. While GAN is faster than the diffusion model, GAN suffers from issues like mode collapse and reduced diversity. This is because GANs rely on single-step inference, whereas diffusion models involve multiple steps of noise addition and denoising, which naturally enhance diversity [2]. By using a diffusion model as a teacher to distill a GAN-like model with distillation loss, we can combine the strengths of both approaches: achieving the good diversity of diffusion models and the fast, one-step inference capability of GAN-like models [3].
>
> Thank you for the insightful question! To address potential confusion, we added a new section “Necessity of Two-Stage Training” in the Supplementary Material.

---

> ### Author Response · Authors · 2024-11-20
> **Response to Reviewer B5oG [Part 4/5]**
>
> > **Q5: When performing distillation, since IPNet only involves a single step of sampling, how is the number of noise-adding and denoising steps for IDE-Net determined?**
>
> **A5** Thank you for your interest in the training details!
>
> **1)** We summarized the training parameters including sampling steps and training weights below:
>
> | **Model Type**           | **Time Step**  | **Loss and Weights**                                                                            | **Training Batch Size** | **Training Steps** |
> |---------------------------|----------------|-------------------------------------------------------------------------------------------------|--------------------------|--------------------|
> | **IDE-Net**               | [0, 999]      | $L_{dm}$ + 0.7 · $L_{aid}$                                                                             | 256                      | 40k                |
> | **IPNet (High time step)**| [400, 800]    | $L_{adv}$ + 1 · $L_{distill}$ + 0 · $L_{triplet}$                                                          | 256                      | 25k                |
> | **IPNet (Middle time step)** | [200, 400]    | $L_{adv}$ + 0.3 · $L_{distill}$ + 0 · $L_{triplet}$                                                        | 256                     | 15k                 |
> | **IPNet (Low time step)** | [150, 200]    | $L_{adv}$ + 0.3 · $L_{distill}$ + 1 · $L_{triplet}$                                                        | 2048                     | 1k                 |
>
> **2)** How is the number of noise-adding and denoising steps for IDE-Net determined?
>
> The student model training is more complicated, which can be counted as three stages based on the noise-adding and denoising time steps:
>
> - **First Stage (High Time Step)**:
> In this stage, the teacher model samples time steps within the range [400, 800]. The loss function in this stage combines Adversarial Loss and Identity Distillation Loss (stochastic noise sampling). The primary goal is to distill the overall structure and pose alignment. The weight of Identity Distillation Loss is set to 1 because the student model is just starting training, the discriminator is weak giving noisy feedback, and thus requires stronger supervision from the teacher. If the weight is set to too small, like 0, the training of the student model is likely to collapse at an early stage, as shown in the paper Figure 6 (c).
>
> - **Second Stage (Middle Time Step)**:
> The time steps in this stage are sampled within the range of [200, 400]. The denoising target of the teacher model in the high sampling step introduces the artifacts, which can be removed in the lower sampling time step in this stage. The pose alignment is further strengthened in this stage. Therefore, the time-step threshold is determined to be 200 by checking the accuracy of spatial alignment and image quality during training. The Identity Distillation Loss (stochastic noise sampling) weight is reduced to 0.3, to weigh more on adversarial loss and improve the image quality of few-step generation.
>
> - **Third Stage (Low Time Step)**:
> In this stage, the time step is sampled within the range of [150, 200] to refine details and enhance style. The loss function is expanded to include: Adversarial Loss, Identity Distillation Loss (DDIM inversion), and Face-Style Enhancing Triplet Loss. Building upon the solid pose alignment for the student model achieved in the high-time-step and middle-time-step stages, this stage is mainly for refining details and enhancing image quality and style: First, the Identity Distillation Loss (DDIM inversion) allows fine-grained distillation, which improves the text fidelity and image quality, compared to Identity Distillation Loss (stochastic noise sampling) in the previous stages; Second, the Face-Style Enhancing Triplet Loss added in this stage strengthens style consistency while preserving identity. A large batch size of 2048 is used to incorporate a greater variety of facial styles within each batch, to stabilize training with the Face-Style Enhancing Triplet Loss. Therefore, the training steps of IPNet are reduced accordingly due to the large batch size.
>
>
> **3)** To provide more clarity, we added a new section “Training Parameter” in the Supplementary Material. Please feel free to reach out if you have additional questions or wish to discuss any specific aspects further.

---

> ### Author Response · Authors · 2024-11-20
> **Response to Reviewer B5oG [Part 5/5]**
>
> > **Q6: Given that the proposed dataset is paired, why wasn’t diffusion loss, some photometric loss, or the losses used in IDENet, applied when training IPNet? Can using those losses provide a more accurate supervision signal than IDENet’s distillation loss?**
>
> **A6** Thank you for your insightful question!
>
> **1)** Diffusion loss and Annealing Identity Loss used in IDE-Net training are not suitable for one-step IPNet:
> - Diffusion loss is primarily employed for training complete diffusion models, aiming to progressively optimize noise prediction throughout the multi-step sampling process to produce high-quality images. However, one of the key objectives of IPNet is to reduce the number of sampling steps to one.
> - Annealing Identity Loss is designed to gradually reduce the weight of Constant Identity Loss across the multiple denoising steps, enabling a smooth transition to text alignment. However, in IPNet’s one-step inference framework, this progressive adjustment is not applicable. Instead, the Annealing Identity Loss will fall back to Constant Identity Loss, which creates two challenges:
>   - Constant Identity Loss enforces strong identity preservation, making it much harder to incorporate the facial style specified by the conditioning text.
>   - Constant Identity Loss can cause visual inconsistencies between the face and background in the output image, especially when there is a noticeable color disparity between the target and input images.
>
> **2)** Why does the Distillation Loss work?
>
> The distillation loss works by aligning the images generated by the target model with the guidance distribution of the diffusion model, effectively distilling the probabilistic distribution of the multi-step diffusion process into a single-step inference. By leveraging the distillation loss, IPNet can inherit the capabilities of the multi-step IDE-Net, including identity preservation, while achieving one-step inference. [3, 4]
>
> Therefore, based on 1) and 2) above, using those losses cannot provide a more accurate supervision signal than IDENet’s distillation loss when training IPNet.
> ******
>
> **Reference**
>
> [1] Ye, Hu, et al. "Ip-adapter: Text compatible image prompt adapter for text-to-image diffusion models." arXiv preprint arXiv:2308.06721 (2023).
> [2] Dhariwal P, Nichol A. Diffusion models beat gans on image synthesis[J]. Advances in neural information processing systems, 2021, 34: 8780-8794.
> [3] Wang Z, Zheng H, He P, et al. Diffusion-gan: Training gans with diffusion[J]. arXiv preprint arXiv:2206.02262, 2022.
> [4] Poole B, Jain A, Barron J T, et al. Dreamfusion: Text-to-3d using 2d diffusion[J]. arXiv preprint arXiv:2209.14988, 2022.

---

> > ### Comment · Reviewer_B5oG · 2024-11-22
> > **Response to authors**
> >
> > Thanks for the detailed responses, which address some of my concerns. I decided to raise my score to 6.

---

> > > ### Author Response · Authors · 2024-11-22
> > >
> > > Thank you so much for your thoughtful feedback! We truly appreciate your recognition of our work and your score adjustment. We are happy to address any further questions and suggestions~

---

### Official Review · Reviewer_QoMT · 2024-10-30

**Soundness:** 3
**Presentation:** 2
**Contribution:** 2
**Rating:** 6
**Confidence:** 5

**Summary:**

This paper introduces the Instant-Portrait Network (IPNet), a one-step diffusion-based real-time portrait image editing model. The proposed IPNet can address challenges such as identity preservation and editing fidelity. The training process has two stages. The first stage is to train an Identity Enhancement Network for robust identity preservation. The second stage uses a novel Diffusion Multi-Objective Distillation approach to train the IPNet with multiple losses. Extensive comparisons demonstrate that IPNet outperforms existing models in identity consistency, instruction fidelity, and speed.

**Strengths:**

1. This paper constructs a paired dataset for portrait image editing, consisting of 10,000,000 pairs of images at a resolution of 1024x576. This extensive dataset is invaluable for both this paper and future research in the field of portrait editing. Additionally, the supplementary materials provide a detailed description of the dataset construction process.
2. This paper presents significant strengths in its approach to portrait image editing. It achieves one-step instruction-based editing with high precision in identity preservation, accurate execution of editing instructions, and fast inference.

**Weaknesses:**

1. The description of the methodology in this paper lacks clarity and omits important details, which may confuse readers. For instance, in section 3.2.2 on Identity Distillation Loss, there is insufficient explanation regarding how the teacher and student models sample time steps during the training process, as well as how the IPNet is distilled to achieve one-step inference. Providing these details would significantly enhance the readers’ understanding of the proposed method.
2. The paper does not provide a comprehensive explanation of the model and experimental details, lacking important information such as the weights assigned to the different loss functions and specific training details for the two stages. Including this information would greatly enhance the clarity and completeness of the work.
3. This paper confines the task to portrait editing under fixed poses, only altering attributes such as clothing, makeup, and style without the capability to generate new poses or expressions. However, the comparison models, IPAdapter and InstantID, offer more versatile functionalities, including the ability to generate new poses and expressions. While IPNet demonstrates superior performance within the scope of this paper, it’s possible that IPAdapter and InstantID, when fine-tuned on the dataset presented in this paper, could achieve comparable results to IPNet.

**Questions:**

1. This paper introduces a dataset for portrait editing. Will it be open-sourced? Making it publicly available would greatly benefit the development of this research area.
2. In constructing the dataset, the authors used IPAdapter combined with ControlNet. Similarly, Table 1 compares the performance of IPAdapter with ControlNet. Could you clarify whether the base models for IPAdapter and the choices for ControlNet in these two instances are the same? Different base model selections will undoubtedly lead to varying performance outcomes.

---

> ### Author Response · Authors · 2024-11-20
> **Response to Reviewer QoMT [Part 1/4]**
>
> We sincerely thank the reviewer for your insightful comments, valuable feedback, and recognition of our work, particularly for acknowledging:
>
> - We introduce a paired dataset for portrait image editing with 10 million image pairs at 1024x576 resolution.
> - We provide a detailed explanation of the dataset construction process.
> - Our method achieves one-step, instruction-based editing with strong identity preservation.
> - Our method ensures the precise execution of editing instructions.
>
> Based on your feedback and suggestions, we have carefully revised the paper. The following updates and enhancements have been implemented accordingly:
>
> - We added a direct clarification when introducing IP-Control-XL in dataset creation.
> - We added a new section titled “Training Parameter” in the Supplementary Material.
> - We added a new section “Resources and Pre-trained Models”  in the Supplementary Material.
> - We added more details, analysis, and plans about our limitations in the Supplementary Material Section “Limitation of Facial Expression Editing”.
> - We compared the facial expression editing with SOTA models and added it in the Supplementary Material Section “Limitation of Facial Expression Editing”.
>
> Once again, thank you for your detailed and constructive input, which has been instrumental in improving the overall quality of our work. Below, we present detailed responses to each of your questions:

---

> ### Author Response · Authors · 2024-11-20
> **Response to Reviewer QoMT [Part 2/4]**
>
> > **Q1: The description of the methodology in this paper lacks clarity and omits important details, which may confuse readers. For instance, in section 3.2.2 on Identity Distillation Loss, there is insufficient explanation regarding how the teacher and student models sample time steps during the training process, as well as how the IPNet is distilled to achieve one-step inference…**
>
> > **Q2: The paper does not provide a comprehensive explanation of the model and experimental details, lacking important information such as the weights assigned to the different loss functions and specific training details for the two stages. Including this information would greatly enhance...**
>
> **A1 & A2:** Thank you for your interest in the training details! I would like to answer the A1 and A2 questions together from the aspects of sampling time steps, model weights, and other parameters for both IDE-Net model training and IPNet distillation.
>
> **1)** We summarized the training parameters below:
>
>
> | **Model Type**           | **Time Step**  | **Loss and Weights**                                                                            | **Training Batch Size** | **Training Steps** |
> |---------------------------|----------------|-------------------------------------------------------------------------------------------------|--------------------------|--------------------|
> | **IDE-Net**               | [0, 999]      | $L_{dm}$ + 0.7 · $L_{aid}$                                                                             | 256                      | 40k                |
> | **IPNet (High time step)**| [400, 800]    | $L_{adv}$ + 1 · $L_{distill}$ + 0 · $L_{triplet}$                                                          | 256                      | 25k                |
> | **IPNet (Middle time step)** | [200, 400]    | $L_{adv}$ + 0.3 · $L_{distill}$ + 0 · $L_{triplet}$                                                        | 256                     | 15k                 |
> | **IPNet (Low time step)** | [150, 200]    | $L_{adv}$ + 0.3 · $L_{distill}$ + 1 · $L_{triplet}$                                                        | 2048                     | 1k                 |
>
> **2)** How is the teacher model IDE-Net trained?
>
> For IDE-Net training, the time steps are sampled within the range **[0, 999]**. We selected **0.7** as Annealing Identity Loss weight to achieve a good balance between identity preservation and image quality. In addition, we compared different annealing algorithms with different weights in the Supplementary Material “Annealing Algorithm Selection For Annealing Identity Loss”.
>
> **3)** How does the student model IPNet sample time steps during the training process? How is the IPNet distilled to achieve one-step inference?
>
> The student model training is more complicated, which can be counted as three stages based on the noise-adding and denoising time steps:
>
> - **First Stage (High Time Step)**:
> In this stage, the teacher samples time steps [400, 800], using a loss function combining Adversarial Loss and Identity Distillation Loss (with stochastic noise sampling). The Identity Distillation Loss weight is set to 1 to provide stronger teacher supervision, as the student model is in early training and the weak discriminator gives noisy feedback. A lower weight risks early collapse, as shown in Figure 6(c). The goal is to distill structure and pose alignment effectively.
>
> - **Second Stage (Middle Time Step)**:
> In this stage, time steps [200, 400] are sampled, reducing artifacts introduced at higher steps and further strengthening pose alignment. The time-step threshold of 200 is chosen based on spatial alignment and image quality. The Identity Distillation Loss weight is reduced to 0.3 to prioritize Adversarial Loss, enhancing image quality.
>
> - **Third Stage (Low Time Step)**:
> In this stage, the time step is sampled within the range of [150, 200] to refine details and enhance style. The loss function is expanded to include: Adversarial Loss, Identity Distillation Loss (DDIM inversion), and Face-Style Enhancing Triplet Loss. Building upon the solid pose alignment for the student model achieved in the high-time-step and middle-time-step stages, this stage is mainly for refining details and enhancing image quality and style: First, the Identity Distillation Loss (DDIM inversion) allows fine-grained distillation, which improves the text fidelity and image quality, compared to Identity Distillation Loss (stochastic noise sampling) in the previous stages; Second, the Face-Style Enhancing Triplet Loss added in this stage strengthens style consistency while preserving identity.
>
> **4)** To provide more clarity, we have added a new section “Training Parameter” in the Supplementary Material of the rebuttal version of the paper. Please feel free to reach out if you have additional questions or wish to discuss any specific aspects further.

---

> ### Author Response · Authors · 2024-11-20
> **Response to Reviewer QoMT [Part 3/4]**
>
> > **Q3: This paper confines the task to portrait editing under fixed poses, only altering attributes such as clothing, makeup, and style without the capability to generate new poses or expressions. However, the comparison models, IPAdapter and InstantID, offer more versatile functionalities, including the ability to generate new poses and expressions. While IPNet demonstrates superior performance within the scope of this paper, it’s possible that IPAdapter and InstantID, when fine-tuned on the dataset presented in this paper, could achieve comparable results to IPNet.**
>
> **A3:** Thank you for your thoughtful feedback. Below, we address your concerns:
>
> **1)** Why do we edit under fixed poses?
>
> Our focus is on real-time, instruction-based portrait editing, tailored for practical applications such as augmented reality (AR), video communication, and filter-based effects. In these contexts, preserving the subject's original pose and position is essential for maintaining user expectations and application functionality (e.g., background replacement or video enhancement). Altering poses in such scenarios would compromise usability, so our design aligns with these practical constraints.
>
> **2)** Why does the method have limited facial expression editing capabilities?
>
> Facial expression editing was not the primary focus during dataset creation, resulting in a limited number of expression-related image pairs in our dataset. Consequently, our method performs less effectively on expression editing (as shown in paper Figure 10) compared to other features including background, clothing, jewelry, hair, and makeup edits. However, it still surpasses state-of-the-art methods in this domain, as demonstrated in paper Figure 11 with the examples of prompts including "smiling".
>
> To address this limitation, we plan to expand our dataset in future iterations by adding more diverse facial expression image pairs, such as "crying", "laughing", and “angry”. This will significantly enhance the model’s ability to handle facial expression editing. This improvement is theoretically feasible, as the identity preservation loss, Annealing Identity Loss (in paper section 3.1.1) is calculated at the embedding level rather than the pixel level, ensuring identity preservation while allowing adjustments to facial expressions.
>
> **3)** Can fine-tuning IPAdapter or InstantID achieve comparable results to IPNet?
>
> No, fine-tuning existing models like IPAdapter does not achieve comparable face similarity. IPAdapter has significantly worse face similarity scores than our IDE-Net and IPNet. This is because IPAdapter lacks strong identity supervision, relying on cross-attention for feature extraction, unlike our method, which employs Annealing Identity Loss to explicitly preserve the identity of the input image. This is evidenced by that the face similarity performance of the dataset created by IPAdapter, which has already been fine-tuned on our target images, significantly underperforms our IDE-Net due to the effectiveness of Annealing Identity Loss.
> | **Source**   | **FaceNet Score ↑** | **InsightFace Score ↑** |
> |--------------|--------------------|-----------------------|
> | **IDE-Net**  | 0.890             | 0.791                |
> | **Dataset**  | 0.749             | 0.576                |
>
> We hope this explanation clarifies the scope and potential of our method, and we are committed to addressing these limitations in our future work. To avoid confusion per your suggestion, we added more details, analysis, and plans about our limitations in the Supplementary Material Section “Limitation of Facial Expression Editing”. In addition, we compared the facial expression editing with SOTA models and added it to the Supplementary Material Section “Limitation of Facial Expression Editing”.  Please let us know if you have further questions or concerns.

---

> ### Author Response · Authors · 2024-11-20
> **Response to Reviewer QoMT [Part 4/4]**
>
> > **Q4: This paper introduces a dataset for portrait editing. Will it be open-sourced? Making it publicly available would greatly benefit the development of this research area.**
>
> **A4:** Thanks for acknowledging the potential value of our dataset to future research! We will try our best to release the dataset, hoping to benefit the development of this research area.
>
> *********
>
> > **Q5: In constructing the dataset, the authors used IPAdapter combined with ControlNet. Similarly, Table 1 compares the performance of IPAdapter with ControlNet. Could you clarify whether the base models for IPAdapter and the choices for ControlNet in these two instances are the same? Different base model selections will undoubtedly lead to varying performance outcomes.**
>
> **A5:** Yes, we use the same IP-Control-XL (IP-Adapter [1] with ControlNet [2]) for dataset creation and SOTA comparison as referenced in Table 1. However, we fine-tuned the IP-Control-XL on the target image before generating image pairs, while the IP-Control-XL referenced in Table 1 was not. Therefore, the face similarity score of the dataset set is higher than referenced in Table 1.  We provide the link and checkpoint of the IP-Control-XL and IP-Control-1.5 below:
>
> **IP-Control-XL:**
> - IPAdapter (ip-adapter-faceid_sdxl.bin): https://huggingface.co/h94/IP-Adapter-FaceID
> - ControlNet-xl (canny): https://huggingface.co/diffusers/controlnet-canny-sdxl-1.0
>
> **IP-Control-1.5:**
> - IPAdapter (ip-adapter-faceid-plus_sd15.bin): https://huggingface.co/h94/IP-Adapter-FaceID
> - ControlNet-1.5 (control_sd15_canny.pth): https://huggingface.co/lllyasviel/ControlNet
>
> We greatly appreciate your feedback, as it helped us identify the need for a clearer explanation. To clarify it in our paper, we added the statement in the Supplementary Material “Dataset”. In addition, we added a new section “Resources and Pre-trained Models” and provided the details of the resources and pre-trained models used in this work, along with their respective URLs.
>
> *********
>
> **Reference**
>
> [1] Ye, Hu, et al. "Ip-adapter: Text compatible image prompt adapter for text-to-image diffusion models." arXiv preprint arXiv:2308.06721 (2023).
> [2] Zhang L, Rao A, Agrawala M. Adding conditional control to text-to-image diffusion models[C]//Proceedings of the IEEE/CVF International Conference on Computer Vision. 2023: 3836-3847.

---

> ### Author Response · Authors · 2024-11-23
> **Response to Reviewer QoMT**
>
> Thank you once again for your thoughtful feedback. If there are any additional concerns or areas that require further clarification, please don’t hesitate to let us know. We are more than happy to discuss and make further improvements.

---

> > ### Comment · Reviewer_QoMT · 2024-11-24
> >
> > Thanks to your responses, some of my concerns were addressed. The details in the responses about model training and distillation made this paper more clear to me.  All three reviewers questioned the details of the model and the training of the model, especially the distillation part. Although the authors responded and added details in the supplemental material, I think that these details are important and help the reader to understand the methodology more clearly. Maybe these details should presented in the main paper. Overall, I will raise my rating to 6.

---

> > > ### Author Response · Authors · 2024-11-25
> > > **Response to Reviewer QoMT**
> > >
> > > Thank you for your valuable feedback. We appreciate the opportunity to address your concerns. Your suggestion to move the training details into the main paper makes perfect sense. Accordingly, we summarized and highlighted the key points of the training details in Section 4.1, "Experiment Setup (Implementation part)," and provided additional information in the supplementary material due to page constraints. We hope this revised version enhances clarity for readers. Please let us know if you have any further questions or concerns; we would be happy to discuss and make further improvements to our paper!

---

### Official Review · Reviewer_15Pc · 2024-11-03

**Soundness:** 3
**Presentation:** 3
**Contribution:** 3
**Rating:** 8
**Confidence:** 5

**Summary:**

This paper tries to tackle the task of portrait editing with identity preservation and style consistency.  It first trains a full-step diffusion model to produce images with robust identity preservation. Then it trains a one-step model that is distilled from the full-step model to further enhance the style, and identity of the edited images.

**Strengths:**

1. The visual results are impressive. Many details are created, while the identity is well-preserved.
2. The approach design is reasonable. It first generates the identity-preserved image, then distills the model for efficient inference.
3. The experiments are convincing.

**Weaknesses:**

Several limitations can be improved to make this paper better.
1. The dataset creation process is reasonable. However, it is still not very clear to me how this dataset is used. The authors mention that when creating the dataset, they use the AID loss to preserve identity. Then what data is used to train the first stage? Is the data from another model that also uses the AID loss?
2. There are many losses included in the training. How are they tuned? Is there a guideline for the tuning of the hyper-parameters?
3. Using the distilled version of the model usually results in a reduced diversity of styles and texture details. If the diversity can be further evaluated, this work can be more convincing.
4. For the facial expression control, any thoughts on why the current model cannot achieve it and any future plan on how to achieve it?
5. Could you list the source code link you use for IP-Control, which also achieves impressive results?

**Questions:**

Please list the details of all the comparison methods, including which codebase you are using, which model checkpoint, etc.

---

> ### Author Response · Authors · 2024-11-20
> **Response to Reviewer 15Pc [Part 1/4]**
>
> We sincerely thank the reviewer for your insightful comments, valuable feedback, and recognition of our work, particularly for acknowledging:
>
> - The generated images generated by our method demonstrate impressive visual quality, successfully creating intricate details while maintaining the subject's identity.
> - Our methodology is thoughtfully structured.
> - Our experimental results effectively validate the proposed approach, showcasing its practical effectiveness.
>
> Based on your feedback and suggestions, we have carefully revised the paper. The following updates and enhancements have been implemented accordingly:
>
> - We enhanced our writing and clarified details to reduce confusion in the Supplementary Material “Dataset” section.
> - We ran new experiments with mean Intra-LPIPS to compare the image diversity of IDE-Net and IPNet and added a new section “Image Diversity Comparison between IDE-Net and IPNet” in the Supplementary Material.
> - We added more details, analysis, and plans about our limitations in the Supplementary Material Section “Limitation of Facial Expression Editing”.
> - We compared the facial expression editing with SOTA models and added it to the Supplementary Material Section “Limitation of Facial Expression Editing”.
> - We added a new section “Training Parameter” in the Supplementary Material.
> - We added a new section “Resources and Pre-trained Models” in the Supplementary Material.
>
> Once again, thank you for your detailed and constructive input, which has been instrumental in improving the overall quality of our work. Below, we present detailed responses to each of your questions:
>
> *********
>
> > **Q1: “The dataset creation process is reasonable. However, it is still not very clear to me how this dataset is used. The authors mention that when creating the dataset, they use the AID loss to preserve identity. Then what data is used to train the first stage? Is the data from another model that also uses the AID loss?”**
>
> **A1:** Thank you for pointing out the confusion! The AID loss is not used during the dataset creation process to preserve identity. Instead, **the AID loss is only employed during the training of IDE-Net. IDE-Net was trained on our created dataset**. While the identity preservation between the input and output images in the dataset we created remains suboptimal, the IDE-Net model significantly improves identity preservation due to the application of the AID loss. This improvement is evident when comparing the identity preservation performance of the dataset and IDE-Net, as shown in the following results:
>
> | **Source**   | **FaceNet Score ↑** | **InsightFace Score ↑** |
> |--------------|--------------------|-----------------------|
> | **IDE-Net**  | 0.890             | 0.791                |
> | **Dataset**  | 0.749             | 0.576                |
>
>
> The confusion may have arisen from the statement in Supplementary Material “Image Pair Generation” section: "However, identity preservation between input and output is still insufficient. This issue is effectively addressed by introducing a novel identity loss in Section 3.1.1."
>
> To clarify, we have updated the statement to "However, identity preservation between the input image and the target image is still insufficient. This issue is effectively addressed, and Face-Similarity scores are significantly improved during the IDE-Net model training stage by introducing a novel identity loss, as described in Section 3.1.1.”
>
> We hope this revision and clarification resolve any confusion. Please feel free to reach out with further questions or suggestions about this.

---

> ### Author Response · Authors · 2024-11-20
> **Response to Reviewer 15Pc [Part 2/4]**
>
> > **Q2: There are many losses included in the training. How are they tuned? Is there a guideline for the tuning of the hyper-parameters?**
>
> **A2:** Thank you for your interest in the training details!
>
> **1)** We summarized the training parameters below:
>
> | **Model Type**           | **Time Step**  | **Loss and Weights**                                                                            | **Training Batch Size** | **Training Steps** |
> |---------------------------|----------------|-------------------------------------------------------------------------------------------------|--------------------------|--------------------|
> | **IDE-Net**               | [0, 999]      | $L_{dm}$ + 0.7 · $L_{aid}$                                                                             | 256                      | 40k                |
> | **IPNet (High time step)**| [400, 800]    | $L_{adv}$ + 1 · $L_{distill}$ + 0 · $L_{triplet}$                                                          | 256                      | 25k                |
> | **IPNet (Middle time step)** | [200, 400]    | $L_{adv}$ + 0.3 · $L_{distill}$ + 0 · $L_{triplet}$                                                        | 256                     | 15k                 |
> | **IPNet (Low time step)** | [150, 200]    | $L_{adv}$ + 0.3 · $L_{distill}$ + 1 · $L_{triplet}$                                                        | 2048                     | 1k                 |
>
>
> **2)** More explanation about teacher model IDE-Net training
>
> For the teacher model training, the time steps are sampled within the range [0, 999]. We selected 0.7 as Annealing Identity Loss weight to achieve a good balance between identity preservation and image quality. In addition, we compared different annealing algorithms with different weights in the Supplementary Material “Annealing Algorithm Selection For Annealing Identity Loss” section.
>
> **3)** More explanation about student model IPNet training
>
> The student model distillation can be counted as two stages based on the noise-adding and denoising time steps:
>
> The student model training is more complicated, which can be counted as three stages based on the noise-adding and denoising time steps:
>
> - **First Stage (High Time Step)**:
> In this stage, the teacher model samples time steps within the range [400, 800]. The loss function in this stage combines Adversarial Loss and Identity Distillation Loss (stochastic noise sampling). The primary goal is to distill the overall structure and pose alignment. The weight of Identity Distillation Loss is set to 1 because the student model is just starting training, the discriminator is weak giving noisy feedback, and thus requires stronger supervision from the teacher. If the weight is set to too small, like 0, the training of the student model is likely to collapse at an early stage, as shown in the paper Figure 6 (c).
>
> - **Second Stage (Middle Time Step)**:
> The time steps in this stage are sampled within the range of [200, 400]. The denoising target of the teacher model in the high sampling step introduces the artifacts, which can be removed in the lower sampling time step in this stage. The pose alignment is further strengthened in this stage. Therefore, the time-step threshold is determined to be 200 by checking the accuracy of spatial alignment and image quality during training. The Identity Distillation Loss (stochastic noise sampling) weight is reduced to 0.3, to weigh more on adversarial loss and improve the image quality of few-step generation.
>
> - **Third Stage (Low Time Step)**:
> In this stage, the time step is sampled within the range of [150, 200] to refine details and enhance style. The loss function is expanded to include: Adversarial Loss, Identity Distillation Loss (DDIM inversion), and Face-Style Enhancing Triplet Loss. Building upon the solid pose alignment for the student model achieved in the high-time-step and middle-time-step stages, this stage is mainly for refining details and enhancing image quality and style: First, the Identity Distillation Loss (DDIM inversion) allows fine-grained distillation, which improves the text fidelity and image quality, compared to Identity Distillation Loss (stochastic noise sampling) in the previous stages; Second, the Face-Style Enhancing Triplet Loss added in this stage strengthens style consistency while preserving identity. A large batch size of 2048 is used to incorporate a greater variety of facial styles within each batch, to stabilize training with the Face-Style Enhancing Triplet Loss. Therefore, the training steps of IPNet are reduced accordingly due to the large batch size.
>
>
> **4)** To provide more clarity, we have added a new section “Training Parameter” in the Supplementary Material of the rebuttal version of the paper. Please feel free to reach out if you have additional questions or wish to discuss any specific aspects further.

---

> ### Author Response · Authors · 2024-11-20
> **Response to Reviewer 15Pc [Part 3/4]**
>
> > **Q3: Using the distilled version of the model usually results in a reduced diversity of styles and texture details. If the diversity can be further evaluated, this work can be more convincing.**
>
> **A3:** Thank you so much for the valuable feedback! Intra-LPIPS (Intra-Learned Perceptual Image Patch Similarity) is a metric for evaluating the diversity of generated images [1]. To compute Intra-LPIPS for a model, we generate multiple images from a single portrait input using different instruction prompts and calculate the pairwise LPIPS among the generated images. For multiple portrait inputs, the mean Intra-LPIPS is computed by averaging the Intra-LPIPS values across all inputs. The mean Intra-LPIPS scores are below:
>
> | **Metric**            | **IDE-Net** | **IPNet** |
> |:----------------------|:-----------:|----------:|
> | Mean Intra-LPIPS Score | 0.571       | 0.589    |
>
>
> **The mean Intra-LPIPS score of the student model IPNet is higher than the teacher model IDE-Net**, because we applied multiple loss functions during the distillation process. According to paper Table 2, IPNet outperforms IDE-Net in text fidelity (including facial style) and image quality, although it shows a slight regression in face similarity. The improved style fidelity but weaker identity preservation indicates that, for the same input image and different text prompts, the variations among the images generated by the IPNet are greater than those produced by the IDE-Net. As a result, the student model IPNet achieves a higher Intra-LPIPS score.
>
> This outcome may seem counterintuitive compared to typical distillation scenarios, where the student model is expected to mimic the teacher more closely and thus worse diversity. However, this “counterintuitive” finding underscores the superiority of our Multi-Objective Distillation approach, which enables the student model to achieve a more balanced and better diversity.
> Thanks again for your insightful suggestion. We have added a new section titled “Image Diversity Comparison between IDE-Net and IPNet” in the Supplementary Material of the rebuttal version of the paper.
>
> *********
>
>
> > **Q4: For the facial expression control, any thoughts on why the current model cannot achieve it and any future plan on how to achieve it?**
>
> **A4:** Thanks for your the the thoughful question!
>
> **1)** Why the current model cannot achieve it?
>
> Facial expression editing was not the primary focus during dataset creation, resulting in a limited number of expression-related image pairs in our dataset. Consequently, our method performs less effectively on expression editing (as shown in paper Figure 10) compared to other features including background, clothing, jewelry, hair, and makeup edits. However, it still surpasses state-of-the-art methods in this domain, as demonstrated in paper Figure 11 with the examples of prompts including "smiling".
>
> **2)** To address this limitation, we plan to expand our dataset in future iterations by adding more diverse facial expression image pairs, such as "crying", "laughing", and “angry”. This will significantly enhance the model’s ability to handle facial expression editing. This improvement is theoretically feasible, as the identity preservation loss, Annealing Identity Loss (in paper section 3.1.1) is calculated at the embedding level rather than the pixel level, ensuring identity preservation while allowing adjustments to facial expressions.
>
> **3)** We hope this explanation clarifies the limitations and potential of our method, and we are committed to addressing these limitations in our future work. To avoid confusion per your suggestion, we added more details, analysis, and plans about our limitations in the Supplementary Material Section “Limitation of Facial Expression Editing”. In addition, we compared the facial expression editing with SOTA models and added it to the Supplementary Material Section “Limitation of Facial Expression Editing”.  Please let us know if you have further questions or concerns.

---

> ### Author Response · Authors · 2024-11-20
> **Response to Reviewer 15Pc [Part 4/4]**
>
> > **Q5: Could you list the source code link you use for IP-Control, which also achieves impressive results?**
> > **Q6: Please list the details of all the comparison methods, including which codebase you are using, which model checkpoint, etc.**
>
> **A5 & A6:** Thanks for your interest in the source code. Here are the links and versions of the IP-Control models (IP-Adapter [2] with ControlNet [3]) and other comparison models.
>
> | Model Name        | Component                    | Link                                                                                      |
> |-------------------|------------------------------|-------------------------------------------------------------------------------------------|
> | **IP-Control-XL** | IPAdapter                    | [ip-adapter-faceid_sdxl.bin](https://huggingface.co/h94/IP-Adapter-FaceID)                |
> |                   | ControlNet-xl (canny)       | [controlnet-canny-sdxl-1.0](https://huggingface.co/diffusers/controlnet-canny-sdxl-1.0)   |
> | **IP-Control-1.5**| IPAdapter                    | [ip-adapter-faceid-plus_sd15.bin](https://huggingface.co/h94/IP-Adapter-FaceID)           |
> |                   | ControlNet-1.5 (canny)      | [control_sd15_canny.pth](https://huggingface.co/lllyasviel/ControlNet)                    |
> | **Instruct-Pix2Pix** |-                             | [sdxl-instructpix2pix-768](https://huggingface.co/diffusers/sdxl-instructpix2pix-768)     |
> | **MagicBrush**    |-                             | [InstructPix2Pix-MagicBrush](https://huggingface.co/osunlp/InstructPix2Pix-MagicBrush)    |
> | **InstantID**     |-                             | [InstantID](https://huggingface.co/InstantX/InstantID)                                    |
>
> We added a new section “Resources and Pre-trained Models” and provided the details of the resources and pre-trained models used in this work, along with their respective URLs. Thanks again for the great question!
>
>
> *********
>
> **References**
>
> [1] Zhu J, Ma H, Chen J, et al. Few-shot image generation via masked discrimination[J]. arXiv preprint arXiv:2210.15194, 2022.
> [2] Ye, Hu, et al. "Ip-adapter: Text compatible image prompt adapter for text-to-image diffusion models." arXiv preprint arXiv:2308.06721 (2023).
> [3] Zhang L, Rao A, Agrawala M. Adding conditional control to text-to-image diffusion models[C]//Proceedings of the IEEE/CVF International Conference on Computer Vision. 2023: 3836-3847.

---

> > ### Comment · Reviewer_15Pc · 2024-11-26
> >
> > Thanks for the response. It resolves some of my concerns. I will keep my score unchanged.

---

> ### Author Response · Authors · 2024-11-27
> **Response to Reviewer 15Pc**
>
> Thank you so much for reviewing our response and for your thoughtful feedback! We’re glad to have addressed your concerns and welcome any further comments or suggestions in the future.

---

### Meta-Review · Area_Chair_5UzW · 2024-12-17

**Metareview:**

The paper introduces a method for editing portrait images that keeps the person's identity clear while making the editing process quicker. It uses a two-step approach with a large dataset of 10 million image pairs to ensure the edits are accurate and preserve identity.

***Strengths:***
- The edited images maintain identity well and look impressive.
- The creation of a new dataset may contribute to the domain
- Techniques like Multi-Objective Distillation and Annealing Identity Loss are new and effective.
- The method allows quick editing without losing much quality.

***Weaknesses:***
- Some parts of the method are not explained clearly, which makes it hard to fully understand. This is critical and should be resolved in the final version.
- The paper doesn't thoroughly compare its results with other models that have more editing features.
- The system can only edit images in fixed poses and doesn't allow for changes in expression or pose.

Reviewers praised the paper and supported acceptance after their concerns were addressed in the rebuttal. The paper adds value to research on image editing with its methods and dataset, and it merits acceptance despite some limitations in scope and method clarity. After careful consideration and discussion, we are pleased to accept this submission.

**Additional Comments On Reviewer Discussion:**

During the rebuttal, the author has provided some more explanation of techinical details and added some more comparisons. The author promised to release the dataset. Two reviewers raised their scores and there became a consensus to accept this paper. However the author still need to take action to refine the presentation for better clarity and ensure the dataset is made available for the community.

---

### Decision · Program_Chairs · 2025-01-22

Accept (Poster)